# Dynamic ubiquitination determines transcriptional activity of the plant immune coactivator NPR1

Michael J Skelly, James J Furniss, Heather Grey, Ka-Wing Wong, Steven H Spoel*

Institute of Molecular Plant Sciences, School of Biological Sciences, University of Edinburgh, Edinburgh, United Kingdom

**Abstract** Activation of systemic acquired resistance in plants is associated with transcriptome reprogramming induced by the unstable coactivator NPR1. Immune-induced ubiquitination and proteasomal degradation of NPR1 are thought to facilitate continuous delivery of active NPR1 to target promoters, thereby maximising gene expression. Because of this potentially costly sacrificial process, we investigated if ubiquitination of NPR1 plays transcriptional roles prior to its proteasomal turnover. Here we show ubiquitination of NPR1 is a progressive event in which initial modification by a Cullin-RING E3 ligase promotes its chromatin association and expression of target genes. Only when polyubiquitination of NPR1 is enhanced by the E4 ligase, UBE4, it is targeted for proteasomal degradation. Conversely, ubiquitin ligase activities are opposed by UBP6/7, two proteasome-associated deubiquitinases that enhance NPR1 longevity. Thus, immune-induced transcriptome reprogramming requires sequential actions of E3 and E4 ligases balanced by opposing deubiquitinases that fine-tune activity of NPR1 without strict requirement for its sacrificial turnover.

DOI: https://doi.org/10.7554/eLife.47005.001

*For correspondence:
steven.spoel@ed.ac.uk

**Competing interests:** The authors declare that no competing interests exist.

## Introduction

Immune responses must be tightly controlled to provide appropriate, efficient and timely resistance to pathogenic threats. A major hallmark of eukaryotic immune responses is dramatic reprogramming of the transcriptome to prioritise defences over other cellular functions. In plants transcriptional reprogramming is largely orchestrated by the immune hormone salicylic acid (SA) that accumulates upon recognition of biotrophic pathogens. SA not only induces resistance in infected local tissues, it is also required for establishment of systemic acquired resistance (SAR), a form of induced resistance with broad-spectrum effectiveness that is long-lasting and protects the entire plant from future pathogen attack (*Spoel and Dong, 2012*). Establishment of SAR and associated transcriptome reprogramming are mediated by the transcriptional coactivator NPR1 (nonexpressor of pathogenesis-related (*PR*) genes 1). The majority of SA-induced genes are NPR1 dependent, indicating NPR1 is a master regulator of plant immunity (*Wang et al., 2006*). Consequently, loss of NPR1 function results in severely immune-compromised plants unable to activate SAR.

Since NPR1 exerts its activity in the nucleus (*Kinkema et al., 2000*), controlling its nuclear entry provides a means to prevent spurious activation of immune responses. Indeed, in resting cells NPR1 is sequestered in the cytoplasm as a large redox-sensitive oligomer that is formed by intermolecular disulphide linkages between conserved cysteine residues (*Mou et al., 2003*). NPR1 monomers that escape oligomerization and enter the nucleus are ubiquitinated by a Cullin-RING Ligase 3 (CRL3), a modular E3 ubiquitin ligase, resulting in their degradation by the 26S proteasome (*Spoel et al., 2009*). Importantly, constitutive clearance of NPR1 from nuclei of resting cells by concerted action of

**eLife digest** Plant diseases cause devastating crop losses around the world and threaten the food supply of millions of people. Over time, plants have developed various mechanisms for fighting off infections caused by pests and other pathogens such as viruses and bacteria. When plants become infected they kick their immune system into action by rapidly switching on and off certain genes. They do this by activating the protein NPR1 which regulates the plant's immune genes. NPR1 is essential for fighting off infections and plants that do not have this protein are highly susceptible to disease.

Peculiarly, once the plant has detected an infection it builds resistance by destroying the NPR1 protein. A previous study suggested that plants do this to replace old 'inactive' NPR1 with newer versions that can activate the genes needed to stop the disease developing. This process, however, requires a lot of energy that could be re-directed to other aspects of the immune response. Now, Skelly et al. – including one of the researchers involved in the previous study – have explored whether there may be other reasons for why plants destroy the NPR1 protein.

Plant cells target NPR1 for destruction by repeatedly tagging it with molecules called ubiquitin to form ubiquitin chains. The length of these chains determines whether a protein is stable and ready for action, or whether it is ready to be destroyed. In experiments with a commonly studied plant known as *Arabidopsis thaliana,* Skelly et al. found that the length of ubiquitin chains attached to the NPR1 protein could fine-tune its level of activity: short ubiquitin chains activate NPR1, while longer chains lead to its destruction and shut down the protein. This suggests that the steps leading to the destruction of NPR1 regulate the immune genes needed to fight off disease.

This work has uncovered important new components of how plants defend themselves from infection. If these findings translate to crop plants they could inform future agricultural strategies for enhancing the plant's own defences to increase crop yields, which would provide more food for a rapidly growing population.

DOI: https://doi.org/10.7554/eLife.47005.002

CRL3 and the proteasome is necessary to prevent untimely activation of its target genes and associated autoimmunity.

Upon activation of SAR, NPR1 is subject to an array of post-translational modifications. A combination of alterations in redox-based modifications, phosphorylation and SUMOylation of NPR1 result in the formation of a transactivation complex that induces the transcription of immune-responsive target genes (*Skelly et al., 2016*; *Withers and Dong, 2016*). Subsequent to these post-translational control points, NPR1 becomes phosphorylated at Ser11 and Ser15, which surprisingly results in recruitment of CRL3 followed by its degradation (*Spoel et al., 2009*). Pharmacological inhibition of the proteasome, genetic mutation of CRL3, and mutation of Ser11/15 all stabilised NPR1 protein, yet impaired the expression of SA-induced NPR1 target genes (*Spoel et al., 2009*). These findings indicate that paradoxically, ubiquitination and degradation of NPR1 are required for the full expression of its target genes. We previously proposed a proteolysis-coupled transcription model in which activation of target gene transcription results in NPR1 being marked as 'spent' by Ser11/15 phosphorylation (*Spoel et al., 2009*). SUMOylation of NPR1 was required for Ser11/15 phosphorylation and facilitates its interaction with other transcriptional activators (*Saleh et al., 2015*), suggesting that NPR1 becomes inactivated only after it has initiated gene transcription. Removal of inactive NPR1 from target promoters may be necessary to allow binding of new active NPR1 protein that can reinitiate transcription, thereby correlating the rate of NPR1 turnover to the level of target gene expression (*Spoel et al., 2009*). This type of transcriptional control by unstable (co)activators has also been reported in other eukaryotes, including for key transcriptional regulators such as the nutrient sensor GCN4 in yeast and the estrogen receptor ERα as well as oncogenic cMyc and SRC-3 activators in humans (*Kim et al., 2003*; *Lipford et al., 2005*; *Métivier et al., 2003*; *Reid et al., 2003*; *von der Lehr et al., 2003*; *Wu et al., 2007*). This suggests that the use of unstable transcriptional (co)activators may be an evolutionary conserved mechanism for fine-tuning gene expression (*Geng et al., 2012*; *Kodadek et al., 2006*).

While transcription-coupled degradation of unstable (co)activators is an attractive model for controlling transcriptional outputs in eukaryotes, it is potentially also a costly sacrificial process. Therefore we explored the alternative possibility that prior to degradation, ubiquitination itself might act as a transcriptional signal. As chains of four or more ubiquitin molecules are thought to be necessary for recruitment of most substrates to the proteasome (*Thrower et al., 2000*), it is plausible that processive ubiquitination could provide a window of opportunity for NPR1 to activate its target genes. In this study we demonstrate that the transcriptional activity of NPR1 is controlled by several ubiquitin chain modifying enzymes. Both stepwise ubiquitin chain extension and trimming activities contribute to the regulation of NPR1 target genes and establishment of plant immunity. Our findings imply that in eukaryotes transcriptional outputs of unstable (co)activators may not be fine-tuned by their proteasomal turnover per se but rather by conjugated ubiquitin chains of dynamic variable length.

## Results

### The E4 ligase UBE4 regulates SA- and NPR1-mediated plant immunity

To examine if stepwise ubiquitination of NPR1 plays a role in plant immune responses we examined a potential role for E4 ligases. Unlike E3 ligases, the E4 class do not contribute towards initial ubiquitination of substrates but rather extend pre-existing ubiquitin chains (*Hoppe, 2005*; *Koegl et al., 1999*). In Arabidopsis the E4 ligase UBE4/MUSE3 has been implicated in the degradation of NLR (nucleotide binding and leucine-rich repeat) immune receptors. Mutant *ube4/muse3* plants exhibited enhanced disease resistance but this phenotype could only be explained in part by the increased stability of an NLR receptor (*Huang et al., 2014*). Therefore we investigated if UBE4 is involved in downstream NPR1-dependent immune signalling by acquiring a loss-of-function T-DNA insertion mutant (*Figure 1—figure supplement 1*). Like mutants in CRL3 ligase that fail to degrade NPR1 (*Spoel et al., 2009*), adult *ube4* plants displayed enhanced expression of immune genes in absence of pathogen challenge (*Figure 1A*). In agreement with this, when the potential for enhanced disease resistance was examined by using a high inoculum of *Psm* ES4326, adult *ube4* mutants showed autoimmunity (*Figure 1B*). To establish if these phenotypes were dependent on SA signalling, *ube4* mutant plants were crossed with SA-deficient *ics1* mutants (*Wildermuth et al., 2001*). The constitutive immune gene expression observed in *ube4* was abolished in *ube4 ics1* double mutant plants (*Figure 1C*). Furthermore, a low inoculum dosage of *Psm* ES4326 that does not cause disease in wild-type (WT) and mutant *ube4* plants, did result in bacterial proliferation in mutant *ics1* plants. In agreement with the gene expression data, enhanced susceptibility was maintained in *ube4 ics1* double mutants (*Figure 1D*), indicating the autoimmune phenotype of adult *ube4* plants is completely dependent on SA. Because SA-dependent immunity is largely regulated by the transcription coactivator NPR1 (*Cao et al., 1997*), we crossed *ube4* with *npr1-1* mutant plants. Constitutive immune gene expression in *ube4* plants was abolished in *ube4 npr1* plants (*Figure 1E*) and this double mutant was equally susceptible to a low *Psm* ES4326 inoculum as *npr1* single mutants (*Figure 1F*). Collectively, these data suggest that in unchallenged plants UBE4 suppresses the expression of SA-mediated NPR1 target genes and prevents autoimmunity, conceivably by altering the stability of upstream NLR immune receptors as well as the downstream NPR1 coactivator.

### UBE4 polyubiquitinates NPR1 coactivator and targets it for degradation

Because *ube4* mutant phenotypes resemble those of mutants in CRL3 ligase (*Spoel et al., 2009*), we investigated if UBE4 also controls NPR1 stability in the nucleus. Expression of a YFP-UBE4 fusion protein in *Arabidopsis* protoplasts confirmed it is indeed partly localised to the nucleus (*Figure 2—figure supplement 1A*). We used the protein synthesis inhibitor cycloheximide to examine if UBE4 controls the stability of SA-induced NPR1.Both SA-induced constitutively expressed NPR1-GFP (*Figure 2—figure supplement 1B*) (*Kinkema et al., 2000*) and endogenous NPR1 from WT plants were degraded within a few hours after exposure to cycloheximide (*Figure 2A and B*). By contrast, both proteins were considerably more stable in the *ube4* mutant genetic background. These findings were further confirmed by quantifying the amount of NPR1-GFP or endogenous NPR1 protein remaining after exposure to cycloheximide (*Figure 2—figure supplement 1C and D*). Stabilisation

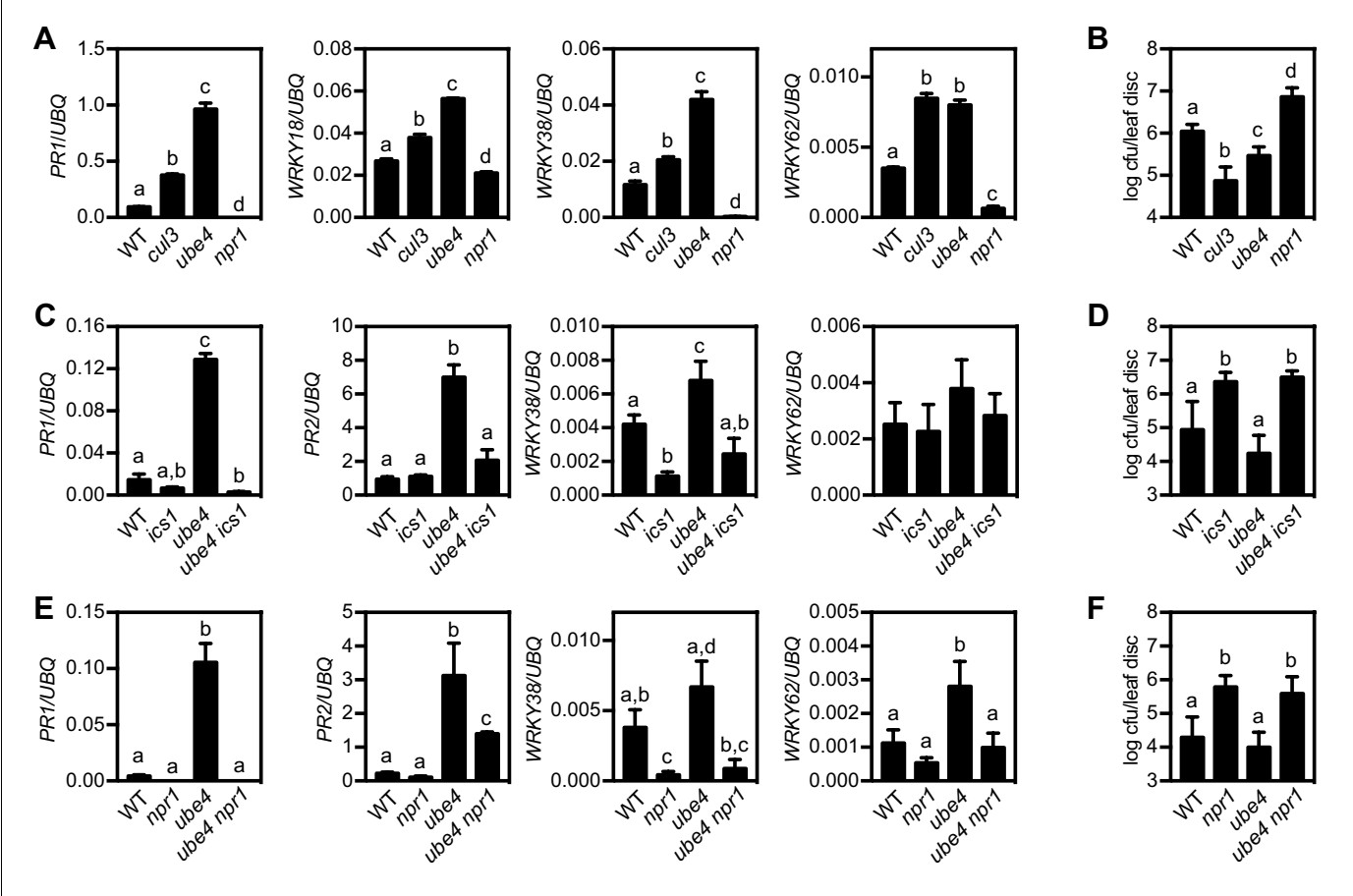

**Figure 1.** The E4 ubiquitin ligase UBE4 regulates SA-mediated plant immunity. (**A**) Expression of NPR1 target genes normalised relative to constitutively expressed *UBQ5* in four-week old plants of the indicated genotypes. Data points represent mean ± SD while letters denote statistically significant differences between samples (Tukey Kramer ANOVA; α = 0.05, n = 3). (**B**) Adult plants were treated with or without 0.5 mM SA 24 hr prior to inoculation with $5 \times 10^6$ colony forming units (cfu)/ml *Psm* ES4326. Leaf discs were analysed for bacterial growth 4 days post-infection (dpi). Error bars represent 95% confidence limits, while letters denote statistically significant differences between samples (Tukey Kramer ANOVA; α = 0.05, n = 8). (**C**) Expression of NPR1 target genes was analysed as in (**A**). (**D**) Adult plants were inoculated with $5 \times 10^5$ cfu/ml *Psm* ES4326 and leaf discs were analysed for bacterial growth at four dpi. Error bars represent 95% confidence limits, while letters denote statistically significant differences between samples (Tukey Kramer ANOVA; α = 0.05, n = 8). (**E**) Basal expression of NPR1 target genes were analysed as in (**A**). (**F**) Adult plants of indicated genotypes were infected and analysed as in (**D**).

DOI: https://doi.org/10.7554/eLife.47005.003

The following figure supplement is available for figure 1:

**Figure supplement 1.** UBE4 knockout.

DOI: https://doi.org/10.7554/eLife.47005.004

of endogenous NPR1 also occurred in the previously described *muse3* mutant allele of *UBE4* (*Figure 2—figure supplement 1E*) (*Huang et al., 2014*). Taken together these results indicate that UBE4 promotes NPR1 degradation. Recruitment of NPR1 to CRL3 for ubiquitination and subsequent degradation requires phosphorylation at residues Ser11 and Ser15 (*Spoel et al., 2009*). Therefore we examined if *ube4* mutants were impaired in NPR1 Ser11/15 phosphorylation. However, Ser11/15 phosphorylation of NPR1-GFP was unaffected by the *ube4* mutation (*Figure 2C*), indicating UBE4 mediates NPR1 turnover downstream of CRL3-mediated ubiquitination.

We then investigated if UBE4 is involved in polyubiquitination of NPR1. Pulldown of polyubiquitinated proteins using tandem-repeated ubiquitin-binding entities (TUBE) (*Hjerpe et al., 2009*) followed by detection of NPR1-GFP, revealed that SA stimulated polyubiquitination of NPR1-GFP (*Figure 2D*). By contrast, SA-induced polyubiquitination of NPR1-GFP was compromised in *ube4* mutants (*Figure 2D*), but ubiquitinated NPR1 was still detected at high-molecular weight. Therefore

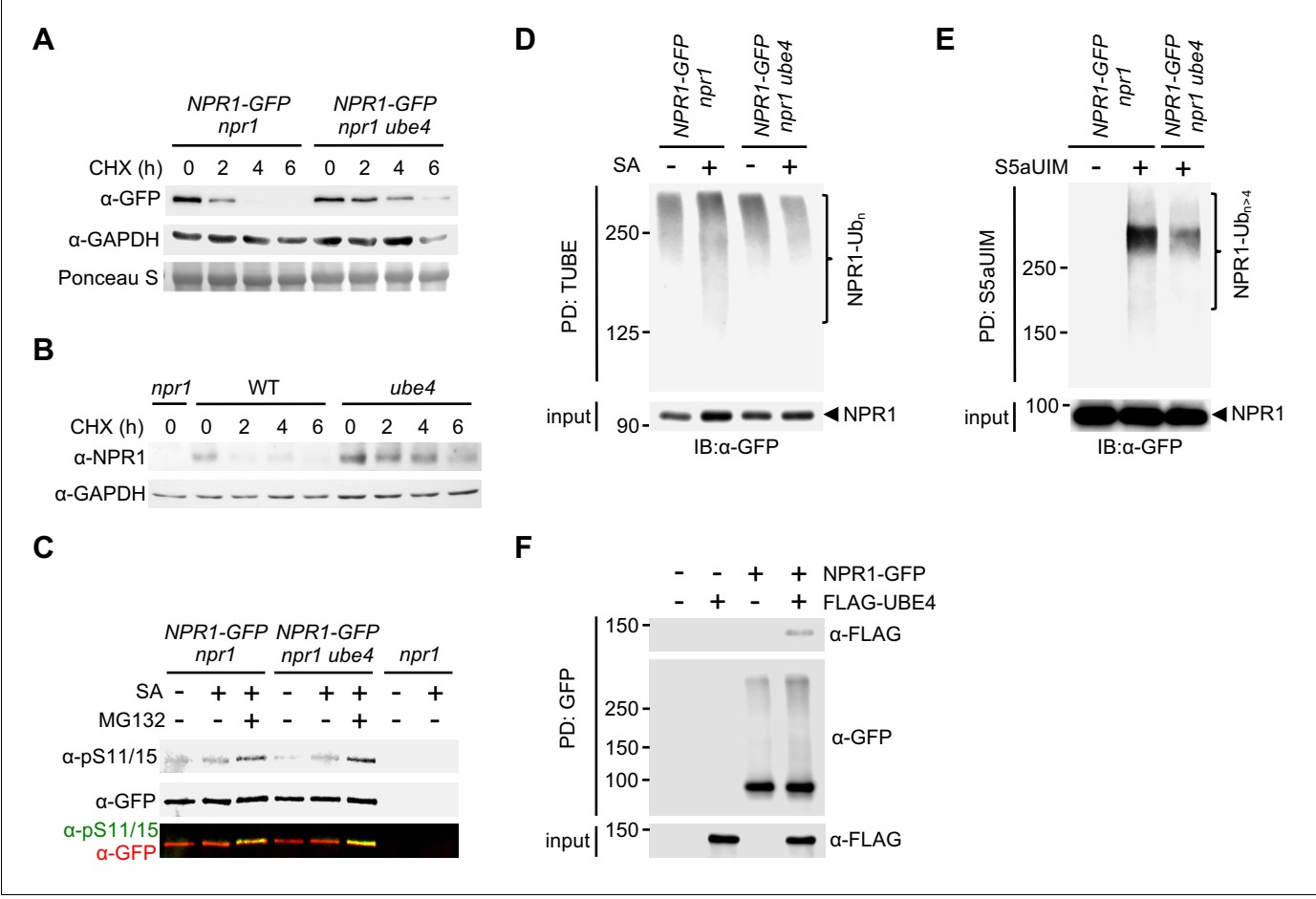

**Figure 2.** UBE4 facilitates polyubiquitination and degradation of NPR1 coactivator. (**A**) Seedlings expressing *35S::NPR1-GFP* in the indicated genetic backgrounds were treated with 0.5 mM SA for 24 hr before addition of 100 µM CHX to inhibit protein synthesis. NPR1-GFP protein levels were monitored by immunoblot analysis, while S5a levels confirmed equal loading. (**B**) Seedlings were treated with 0.5 mM SA for 24 hr before addition of 100 µM CHX. Endogenous NPR1 protein levels were then monitored at the indicated times by immunoblot analysis, while GAPDH levels confirmed equal loading. (**C**) Seedlings expressing *35S::NPR1-GFP* in the indicated genetic backgrounds were pre-treated with 0.5 mM SA for 2 hr followed by addition of vehicle (DMSO) or 100 µM MG132 for an additional 4 hr. Phosphorylated Ser11/15 (pS11/15) and total NPR1-GFP levels were then determined by immunoblotting. (**D**) Seedlings expressing *35S::NPR1-GFP* in the indicated genetic backgrounds were pre-treated with 0.5 mM SA for 6 hr followed by addition of 100 µM MG132 for an additional 18 hr before ubiquitinated proteins were pulled down using GST-TUBEs. Input and ubiquitinated NPR1-GFP (NPR1-Ub$_n$) were detected by immunoblotting with a GFP antibody. (**E**) Seedlings expressing *35S::NPR1-GFP* in the indicated genetic backgrounds were pre-treated with 0.5 mM SA for 2 hr followed by addition of 100 µM MG132 for an additional 4 hr before ubiquitinated proteins were pulled down (PD) using His$_6$-V5-S5a-UIMs. Unmodified and long-chain polyubiquitinated NPR1-GFP (NPR1-Ub$_{n>4}$) were detected by immunoblotting with GFP antibodies. (**F**) Seedlings expressing *35S::NPR1-GFP* in the indicated genetic backgrounds were treated for 6 hr with 0.5 mM SA followed by addition of 100 µM MG132 for a further 18 hr. Polyubiquitinated NPR1-GFP protein was then purified with GFP-Trap agarose and incubated for 2 hr with in vitro synthesised FLAG-UBE4. NPR1-GFP was detected by immunoblotting with GFP antibodies, while FLAG-UBE4 was detected using FLAG antibodies.

DOI: https://doi.org/10.7554/eLife.47005.005

The following figure supplement is available for figure 2:

**Figure supplement 1.** UBE4 cellular localisation and effect on NPR1 stability.

DOI: https://doi.org/10.7554/eLife.47005.006

we sought to distinguish if in *ube4* mutants, NPR1 was modified by long ubiquitin chains or multiple shorter chains, both of which yield high-molecular weights on SDS-PAGE. Thus, we performed pull down experiments with recombinant S5a ubiquitin interacting motifs (S5aUIM) that preferentially bind chains of four or more ubiquitin molecules (*Deveraux et al., 1994*; *Young et al., 1998*). Compared to plants carrying wild-type *UBE4* alleles, the amount of SA-induced polyubiquitinated NPR1-GFP pulled down with recombinant S5aUIM was strikingly lower in *ube4* mutants (*Figure 2E*),

indicating that UBE4 promotes formation of long ubiquitin chains on NPR1 leading to its proteasomal degradation. To determine if UBE4 may act directly on NPR1, we assessed the ability of UBE4 to physically interact with ubiquitinated NPR1. We isolated ubiquitinated NPR1-GFP from SA and MG132-treated *ube4* mutants and subsequently incubated with recombinant FLAG-UBE4. As shown in *Figure 2F*, NPR1-GFP specifically pulled down FLAG-UBE4, indicating that UBE4 physically binds to polyubiquitinated NPR1 to facilitate ubiquitin chain extension.

## Progressive ubiquitination controls transcriptional activity of NPR1

Because UBE4 enhanced polyubiquitination of NPR1 and controlled its stability (*Figure 2*), we investigated if similar to CRL3 (*Spoel et al., 2009*), it also promotes transcriptional activity of NPR1. In stark contrast to *cul3a cul3b* mutants that were compromised in SA-induced expression of NPR1 target genes, plants carrying two different *ube4* mutant alleles exhibited elevated expression levels that were much higher than in WT (*Figure 3A and B*, *Figure 3—figure supplement 1A*). To explore the effect of UBE4 on the NPR1-dependent transcriptome, we performed RNA Seq on SA-treated WT, *ube4* and *npr1* plants. Among 2612 genes whose expression changed by ≥2 fold in response to SA in WT or *ube4* mutants, 75% were stringently dependent on NPR1 (*i.e.* ≥1.5 fold difference compared to *npr1*) (*Figure 3—source data 1*). We separated these genes into two categories: (1) genes that were regulated by SA in both WT and mutant *ube4* plants, and (2) genes that did not make the ≥2 fold change cut-off in WT but were highly regulated by SA in *ube4* mutants. The majority of SA-induced genes in category 1, including *PR1* and *WRKY* marker genes, received a boost in expression when *UBE4* was knocked out (*Figure 3C*). This positive effect was even clearer for category two genes (*Figure 3C and D*). Similarly, genes suppressed by SA treatment displayed further downregulation in *ube4* mutants compared to WT (*Figure 3C*). By contrast, SA-regulated genes that were not dependent on NPR1 behaved similarly in WT and mutant *ube4* plants (*Figure 3—figure supplement 1B*), suggesting UBE4 exerts its effects predominantly through NPR1. Together these data suggest that in absence of UBE4-mediated long-chain polyubiquitination, NPR1 remains in a highly active transcriptional state.

To understand the opposing effects of CRL3 and UBE4 on transcriptional activity of NPR1, we examined endogenous NPR1 protein levels. Compared to WT plants, SA-induced NPR1 accumulated to elevated levels in both *cul3a/b* and *ube4* mutants (*Figure 3E*). Thus, NPR1 protein levels cannot explain differences in transcriptional output of NPR1. We then examined if changes in polyubiquitin chain length regulate NPR1 association with its target promoters. To that end we performed chromatin immunoprecipitation experiments on plants that constitutively expressed NPR1-GFP, thereby eliminating genotype-dependent differences in NPR1 protein level. Coinciding with elevated *PR1* gene expression, at 8 hr after SA treatment more NPR1-GFP was bound to the *PR1* promoter in *ube4* mutants compared to plants carrying wild-type *UBE4* alleles (*Figure 3F*). This indicates that in absence of long polyubiquitin chains, early occupancy by transcriptionally competent NPR1 is increased at target promoters. We also examined a later time point after SA treatment (24 hr) and found that NPR1-GFP was still associated with the *PR1* promoter in plants expressing wild-type *UBE4*, but not in *ube4* mutants (*Figure 3G*). Nonetheless, *PR1* gene expression remained at elevated levels in these mutants (*Figure 3G*), implying that in absence of long-chain polyubiquitination NPR1 strongly switches on target genes without the need for long-term residency at their promoters.

To investigate if CRL3 and UBE4 act independently or in tandem, we crossed *ube4* single with *cul3a cul3b* double mutants and analysed the expression of NPR1 target genes in the resulting triple mutant. Strikingly, *cul3a cul3b ube4* mutants showed severe developmental defects, including stunted growth and complete sterility (*Figure 3—figure supplement 1C and D*). Very few viable homozygous plants were recovered, perhaps suggesting these two ligases work together and share substrates. Nonetheless we were able to select just enough plants to examine the behaviour of NPR1 target genes. In *cul3a cul3b ube4* mutants the SA-induced expression of several genes, including *PR* genes, was impaired to a similar extend as in *cul3a cul3b* double mutants, indicating that elevated gene expression observed in *ube4* plants is dependent on CRL3 (*Figure 3H and I*). However, a subset of NPR1 target genes (*i.e.* WRKY18, WRKY38, WRKY62) were dramatically upregulated in *cul3a cul3b ube4* mutants to a level higher than in any of the other genotypes. This suggests that in absence of CRL3 and UBE4, these genes were activated through another pathway or were highly responsive to elevated homeostatic levels of NPR1 protein and may therefore not be suitable readouts for this particular epistatic analysis (*Figure 3I*). Regardless of this specific, our broader findings

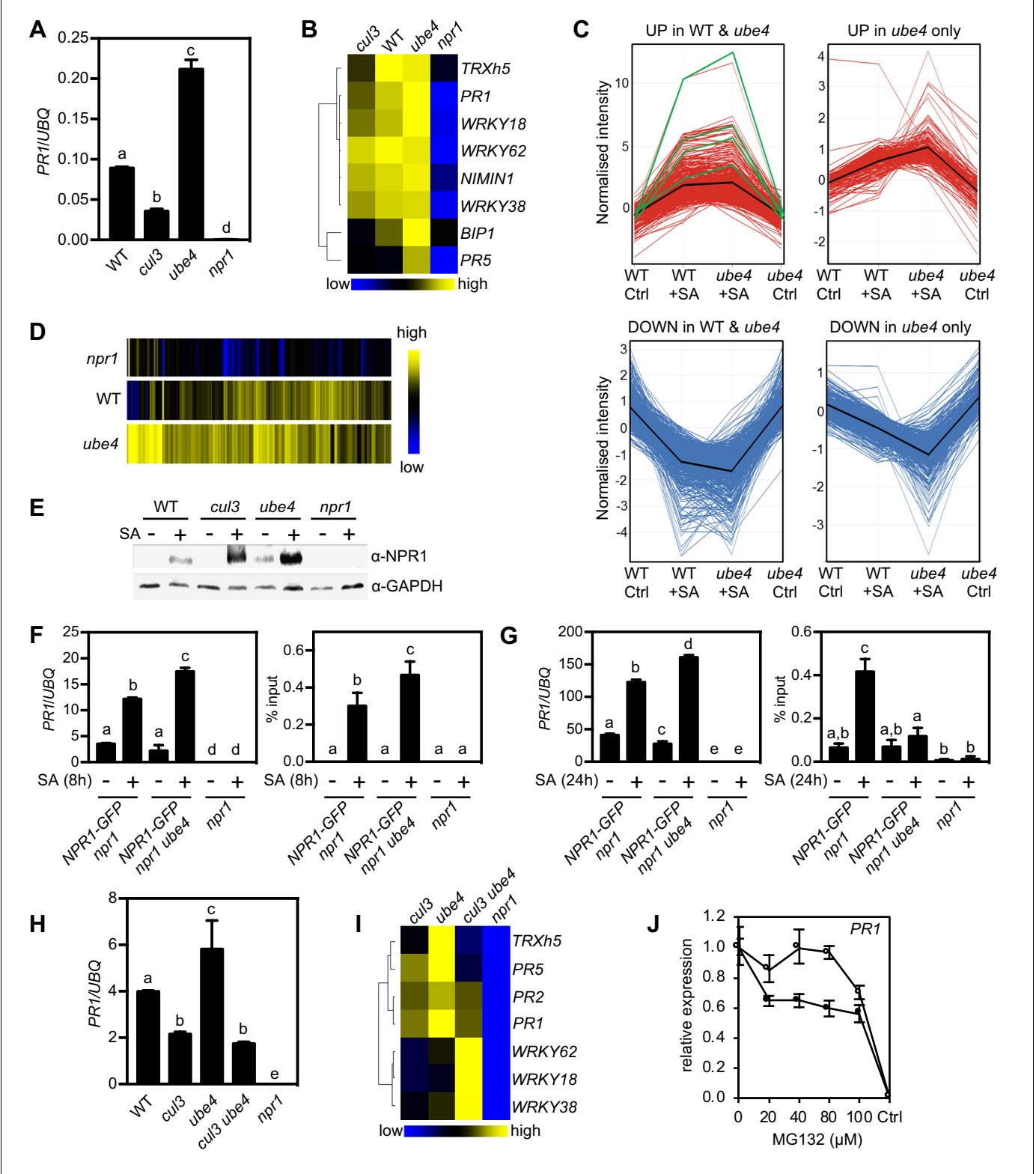

**Figure 3.** Progressive ubiquitination controls transcriptional activity of NPR1. (**A**) WT, *cul3a cul3b* (*cul3*), *ube4* and *npr1* seedlings were treated with 0.5 mM SA for 6 hr before determining *PR1* gene expression normalised relative to constitutively expressed *UBQ5*. Data points represent mean ± SD while letters denote statistically significant differences between samples (Tukey Kramer ANOVA; α = 0.05, n = 3). (**B**) Heat map of the expression of additional NPR1 target genes analysed as in (**A**). (**C**) Seedlings treated with water (Ctrl) or 0.5 mM SA for 12 hr were analysed by RNA-Seq. Only genes that were induced ≥2 fold by SA in WT and/or *ube4* plants and showed ≥1.5 fold difference in expression in *npr1* mutants are shown (Benjamini Hochberg FDR,
*Figure 3 continued on next page*

*Figure 3 continued*

2-way ANOVA p≤0.05). Graphs indicate genes that are up or down regulated in both WT and *ube4* or only in *ube4*. *PR-1*, *WRKY18*, *WRKY38* and *WRKY62* marker genes are indicated by green lines, whereas mean expression patterns are indicated by black lines. (D) Heat map representation of genes from (C) that were upregulated by SA. (E) WT, *cul3a cul3b* (*cul3*), *ube4* and *npr1* seedlings were treated with water (-) or 0.5 mM SA (+) for 6 hr. Endogenous NPR1 protein levels were monitored by immunoblot analysis, while GAPDH levels confirmed equal loading. (F) Adult plants expressing *35S::NPR1-GFP* in the indicated genetic backgrounds were treated with 0.5 mM SA for 8 hr before analysing either *PR1* gene expression (left panel) or NPR1-GFP binding to the *as-1* motif of the *PR1* promoter (right panel). Mutant *npr1* plants served as a negative control. Data points represent mean ± SD while letters denote statistically significant differences between samples (Tukey Kramer ANOVA; α = 0.05, n = 3). (G) As in (F) except plants were treated with 0.5 mM SA for 24 hr. (H) WT, *cul3a cul3b* (*cul3*) double, *ube4* single, *cul3a cul3b ube4* (*cul3 ube4*) triple and *npr1* single mutant seedlings were treated with 0.5 mM SA for 6 hr and *PR1* gene expression determined by normalising against constitutively expressed *UBQ5*. Data points represent mean ± SD while letters denote statistically significant differences between samples (Tukey Kramer ANOVA; α = 0.05, n = 3). (I) Heat map of the expression of additional NPR1 target genes analysed as in (H). (J) WT (closed circles) and mutant *ube4* (open circles) seedlings expressing *35S:NPR1-GFP* were treated with 0.5 mM SA for 4 hr followed by the addition of indicated concentrations of MG132 for an additional 2 hr. *PR1* gene expression was determined and normalised relative to constitutively expressed *UBQ5*. MG132 treatments as well as a control (Ctrl) that received 4 hr of water treatment followed by the addition of vehicle (DMSO), were plotted relative to maximal SA-induced *PR1* expression. Data points represent mean ± SD (n = 3).

DOI: https://doi.org/10.7554/eLife.47005.007

The following source data and figure supplement are available for figure 3:

**Source data 1.** SA-induced genes in WT, *ube4* and *npr1* plants determined by RNA-Seq.
DOI: https://doi.org/10.7554/eLife.47005.009
**Figure supplement 1.** Progressive ubiquitination controls transcriptional activity of NPR1.
DOI: https://doi.org/10.7554/eLife.47005.008

suggest that CRL3 and UBE4 function sequentially in the stepwise addition or extension of ubiquitin chains on NPR1 but with opposing effects on its transcriptional activity.

We then examined if in *ube4* mutants NPR1 lingered in a highly transcriptional active state that does not require proteasome-mediated turnover. To negate any feedback effects of loss of UBE4 activity on endogenous NPR1 expression, seedlings constitutively expressing NPR1-GFP were treated with SA plus a range of MG132 concentrations. SA-induced *PR1* and *WRKY* gene expression was inhibited by increasing concentrations of MG132 in *NPR1-GFP* (in *npr1*) plants (*Figure 3J* and *Figure 3—figure supplement 1E*). By contrast, the SA-induced expression of these NPR1 target genes was largely unresponsive to MG132 in *ube4* mutants, especially at lower concentrations. Thus, loss of UBE4 largely uncoupled NPR1 target gene expression from proteasome activity, demonstrating the importance of progressive ubiquitination for NPR1 activity. In summary, our findings indicate that initial CRL3-mediated ubiquitination is required for NPR1 to attain its full transcriptional activity, while the stepwise formation of long ubiquitin chains mediated by UBE4 inactivates NPR1 and promotes its degradation by the proteasome.

## Deubiquitinases regulate NPR1-dependent transcription

Trimming or removal of ubiquitin chains is performed by deubiquitinases (DUBs) and may provide another layer of regulation of NPR1 activity. The *Arabidopsis* genome is predicted to encode for at least 65 DUBs (*Vierstra, 2009*; *Yang et al., 2007*) with high likelihood of redundancy among gene families. Therefore identifying candidate genes that potentially regulate NPR1 by genetically screening mutant collections was not feasible. Instead, we used a range of pharmacological broad-spectrum and selective DUB inhibitors and assessed their effect on SA-induced gene expression. The broad-spectrum inhibitors PR-619 (*Altun et al., 2011*) and NSC632839 (*Aleo et al., 2006*) strongly impaired SA-induced gene expression across all NPR1 target genes tested (*Figure 4A*), suggesting that DUB activity is required for their optimal expression. Furthermore, while treatment with PR-619 or NSC632839 did not affect SA-induced transcription of the *NPR1* gene, it depleted NPR1 protein levels (*Figure 4B*). Thus, DUB activity may not only be required for NPR1-dependent gene expression but also for increasing NPR1 stability.

Next we tested more selective inhibitors that more specifically target one or a few DUBs. First we treated WT seedlings with various DUB inhibitors and compared the cellular levels of global ubiquitin conjugates with control-, and MG132-treated seedlings. While NSC632839 and MG132 treatments dramatically enhanced accumulation of ubiquitin conjugates, especially in combination with SA treatment, all other inhibitors had little effect on cellular ubiquitination levels (*Figure 4C*). We

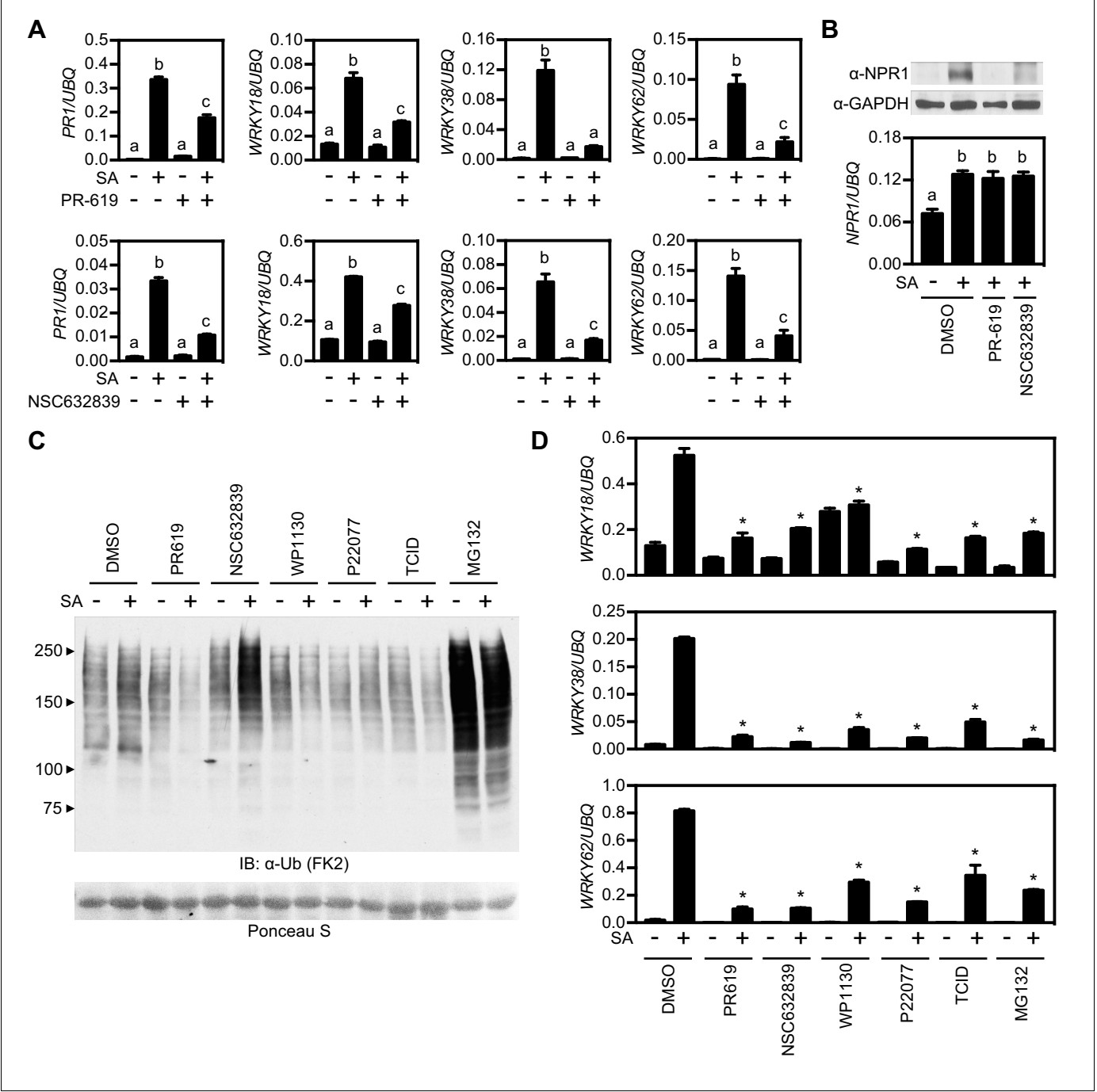

**Figure 4.** Deubiquitinases regulate NPR1-dependent transcription. (A) WT seedlings were treated for 6 hr with either vehicle control (DMSO) or the indicated DUB inhibitors (50 µM) in presence or absence of 0.5 mM SA before analysing the expression of NPR1 target genes. Data points represent mean ± SD while letters denote statistically significant differences between samples (Tukey Kramer ANOVA; α = 0.05, n = 3). (B) WT seedlings were treated as in (A) before endogenous NPR1 and GAPDH (loading control) protein levels were analysed by immunoblotting (top panel). NPR1 gene expression was also analysed from the same samples (bottom panel). Data points represent mean ± SD while letters denote statistically significant differences between samples (Tukey Kramer ANOVA; α = 0.05, n = 3). (C) WT seedlings were treated for 6 hr with vehicle (DMSO) or either the indicated DUB inhibitors (50 µM) or MG132 (100 µM) in presence or absence of 0.5 mM SA before immunoblotting against conjugated ubiquitin (FK2). Ponceau S staining indicated equal loading. (D) WT seedlings were treated as in (C) and NPR1 target gene expression analysed. Data points represent mean ± SD while asterisks denote statistically significant differences between the indicated samples and the DMSO + SA treated sample (Tukey Kramer ANOVA; α = 0.05, n = 3).

DOI: https://doi.org/10.7554/eLife.47005.010

*Figure 4 continued on next page*

*Figure 4 continued*

The following figure supplement is available for figure 4:

**Figure supplement 1.** DUB inhibitors suppress NPR1 target gene expression.

DOI: https://doi.org/10.7554/eLife.47005.011

then examined the effect of these DUB inhibitors on SA-induced gene expression. All inhibitors strongly suppressed SA-induced expression of NPR1 target genes (*Figure 4D*). Furthermore, most inhibitors were effective at low micromolar concentrations and suppressed NPR1 target genes in a dose-dependent manner (*Figure 4—figure supplement 1*). Collectively these data provide a first indication that DUB activity may be crucial for NPR1 stability and efficient activation of SA-induced NPR1 target genes.

## Identification of DUBs that regulate NPR1-dependent transcription

The more selective inhibitors used in experiments described above have been shown to target DUBs in mammalian cells (*Figure 5—figure supplement 1A*) (*Altun et al., 2011*; *Kapuria et al., 2010*; *Liu et al., 2003*). To find potential homologues we used the sequences of these mammalian DUBs to search the *Arabidopsis* genome using BLASTp. The identified *Arabidopsis* DUBs included members of the *ubiquitin-specific protease* (*UBP*) and *ubiquitin C-terminal hydrolase* (*UCH*) multi-gene families (*Figure 5—figure supplement 1A*). We then searched mutant collections to identify T-DNA knockouts for each of these DUBs. *UBP14* knockouts are lethal in *Arabidopsis* (*Doelling et al., 2001*), while no T-DNA insertions were identified for either *UCH1* or *UCH2* in mutant collections of the Col-0 genetic background. Therefore we did not pursue these DUBs further. The DUB inhibitor TCID is thought to target mammalian UCH-L3 for which we identified a single *Arabidopsis* homologue, UCH3. We acquired a T-DNA insertion line that displayed complete knockout of *UCH3* expression (*Figure 5—figure supplement 1B*) and analysed SA-induced NPR1 target gene expression. *Figure 5A* shows that SA-induced *PR1* and *WRKY* gene expression was generally comparable between *uch3* and WT plants, indicating UCH3 is unlikely to play a major role. Next we identified UBP12 and UBP13 as potential plant targets of both WP1130 and P22077 inhibitors (*Figure 5—figure supplement 1A*). Previous research has suggested a role for these two proteins in plant immunity, as *ubp12 ubp13* double knockdown RNAi plants exhibited elevated expression of *PR1* and increased resistance to the virulent pathogen *P. syringae* pv. *tomato* (*Ewan et al., 2011*). Single knockout mutants of *UBP12* and *UBP13* have no observable phenotype and double knockouts are seedling lethal (*Cui et al., 2013*; *Ewan et al., 2011*). However we acquired the *ubp12-2w* allele, previously described as a weak *ubp12 ubp13* double mutant (*Cui et al., 2013*), and analysed this mutant for SA-induced gene expression. Similar to a previous report (*Ewan et al., 2011*), we observed elevated *PR1* expression in *ubp12-2w* plants but other NPR1 target genes were activated to a slightly lesser extent as in WT (*Figure 5B*). This phenotype does not explain the suppressive effects we observed with pharmacological DUB inhibitors. Finally, we acquired T-DNA knockout lines for the mammalian USP14 homologues, UBP6 and UPB7 that are potentially targeted by the WP1130 inhibitor (*Figure 5—figure supplement 1A and C*). SA-induced expression of *PR1* was slightly lower in these mutants but *WRKY* gene expression was largely comparable to WT plants (*Figure 5C*). Since UBP6 and UBP7 are close homologues (*Figure 5—figure supplement 2A and B*), we generated *ubp6 ubp7* double knockout mutants (*Figure 5—figure supplement 1C*) that were viable and showed no observable developmental phenotypes. However, *ubp6 ubp7* mutants were impaired in activation of SA-induced gene expression (*Figure 5D*). This indicates that UBP6 and UBP7 are functionally redundant and required for NPR1 target gene expression.

## UBP6 is a proteasome-associated DUB that deubiquitinates NPR1

Human USP14 and its yeast homologue Ubp6 have both been shown to associate with the 26S proteasome, which is necessary for their activity (*Borodovsky et al., 2001*; *Leggett et al., 2002*). We tested if this is also the case for *Arabidopsis* UBP6 by constitutively expressing FLAG-tagged UBP6 in the *ubp6 ubp7* double mutant background followed by co-immunoprecipitation experiments. The proteasomal subunits S5a and RPN6 both co-immunoprecipitated with FLAG-tagged UBP6 (*Figure 6A*), indicating UBP6 is also a proteasome-associated DUB in plants.

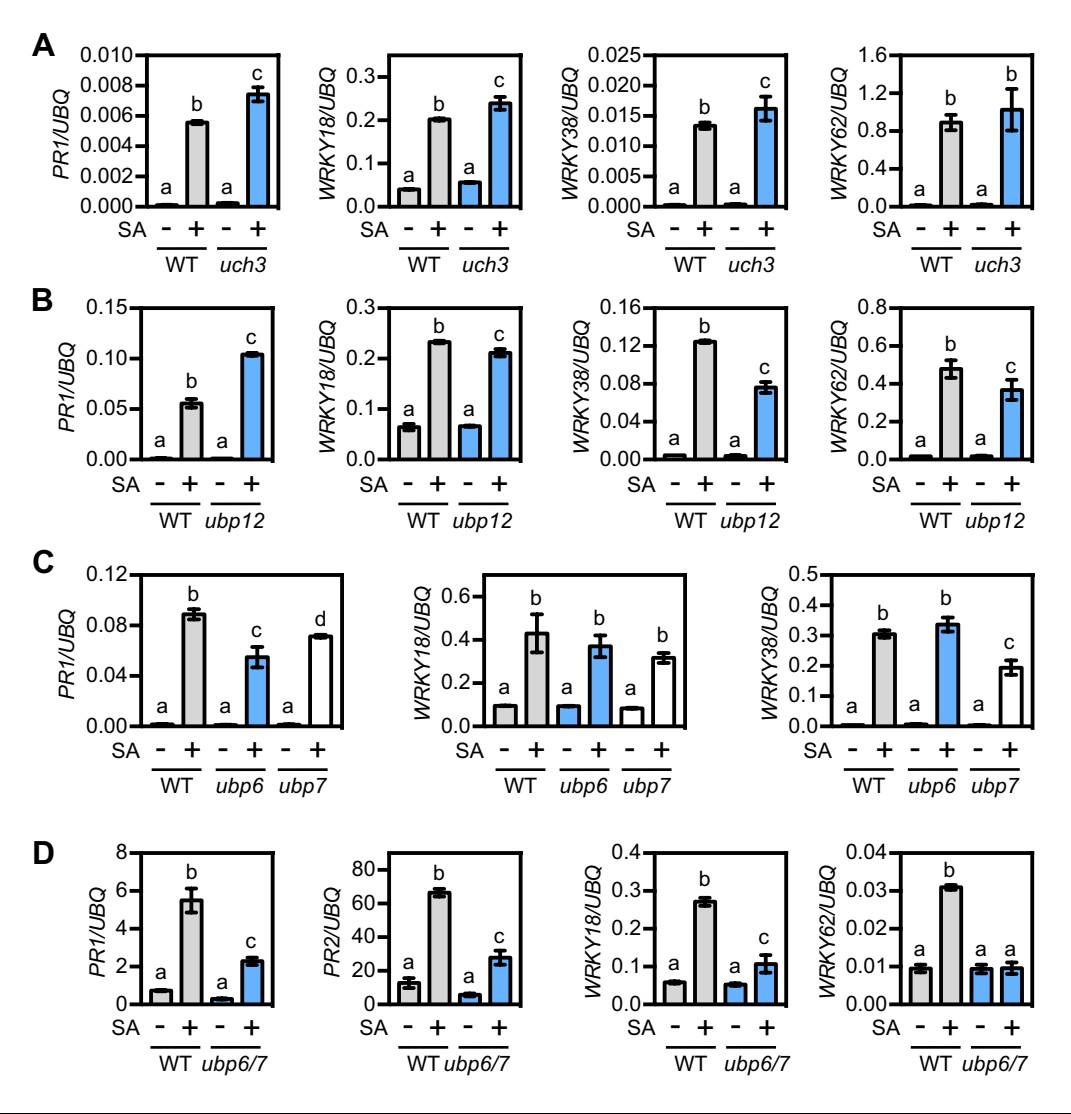

**Figure 5.** UBP6 and UBP7 deubiquitinases are required for SA-induced expression of NPR1 target genes. (A) WT and *uch3-1* seedlings were treated for 6 hr with 0.5 mM SA followed by analysis of NPR1 target gene expression. Data points represent mean ± SD while letters denote statistically significant differences between samples (Tukey Kramer ANOVA; α = 0.05, n = 3). (B) WT and *ubp-12–2* w seedlings were treated and analysed as in (A). (C) WT, *ubp6-1* and *ubp7-1* plants were treated with 0.5 mM SA for 24 hr before analysis of NPR1 target gene expression. Data points represent mean ± SD while letters denote statistically significant differences between samples (Tukey Kramer ANOVA; α = 0.05, n = 3). (D) WT and *ubp6-1 ubp7-1* double mutant plants were treated and analysed as in (C).

DOI: https://doi.org/10.7554/eLife.47005.012

The following figure supplements are available for figure 5:

**Figure supplement 1.** DUB inhibitor targets in Arabidopsis.
DOI: https://doi.org/10.7554/eLife.47005.013

**Figure supplement 2.** Domain structure and sequence of UBP6 and UBP7.
DOI: https://doi.org/10.7554/eLife.47005.014

Next we examined if UBP6 exhibits typical DUB activity. We produced recombinant T7-tagged UBP6 and incubated it with HA-tagged ubiquitin vinyl sulfone (HA-UbVS), an ubiquitin mimic that cannot be hydrolysed upon irreversible binding to DUB active sites (*Borodovsky et al., 2001*). HA-UbVS readily labelled T7-UBP6 but only upon addition of 26S proteasomes (*Figure 6B*), indicating

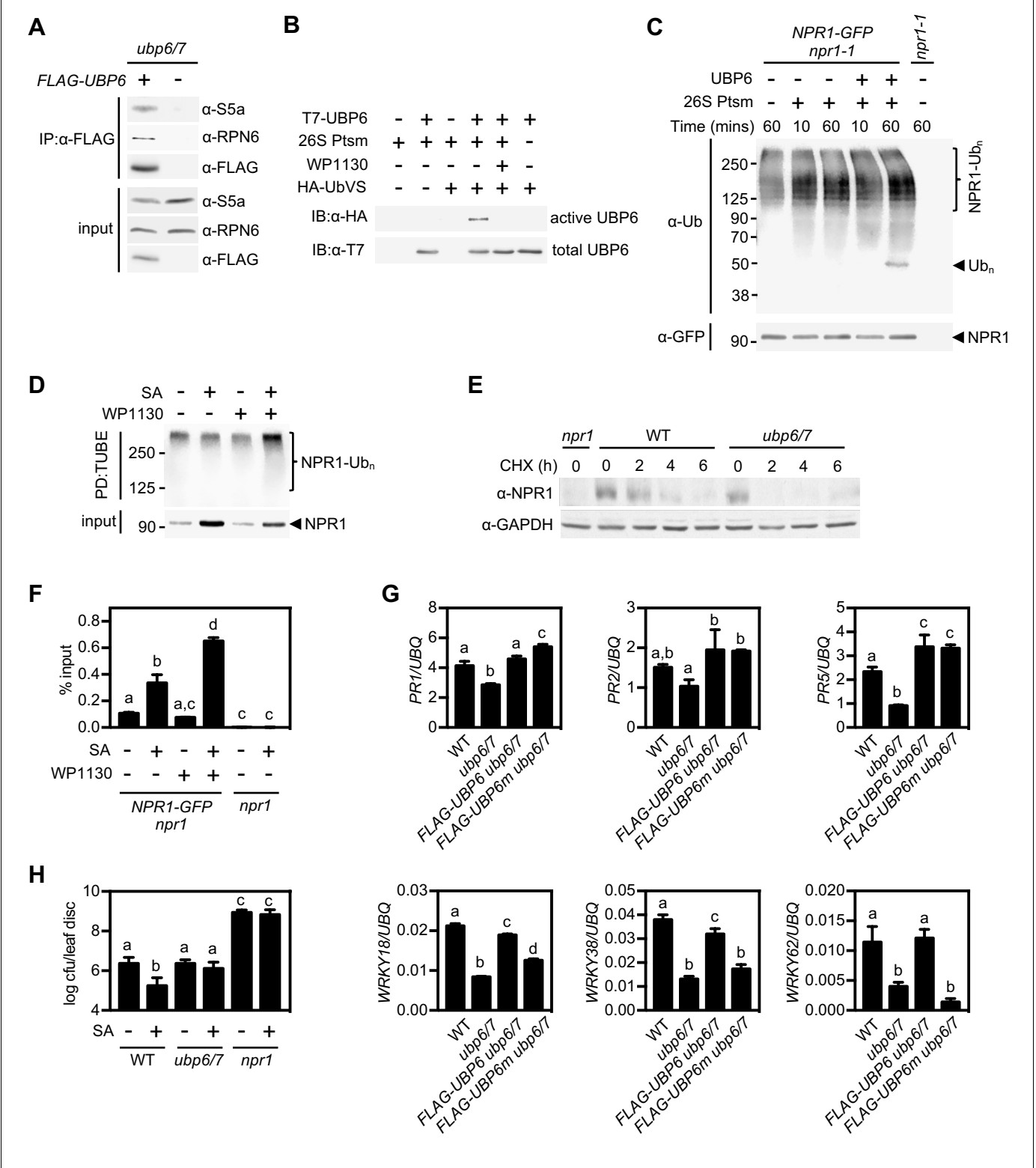

**Figure 6.** Deubiquitination by UBP6/7 regulates transcriptional activity of NPR1. (**A**) FLAG-UBP6 was immunoprecipitated (IP) from *ubp6 ubp7* plants transformed with or without *35S::FLAG-UBP6*. Co-immunoprecipitates were analysed by immunoblotting against FLAG as well as the proteasome subunits S5a and RPN6. Input protein levels are shown in the bottom panel. (**B**) Purified recombinant His$_6$-T7-UBP6 was preincubated with or without WP1130 and 26S proteasomes before labelling with HA-UbVS. Immunoblotting with HA antibodies detected active, labelled UBP6 while immunoblotting with T7 antibodies detected total levels of UBP6. (**C**) *35S::NPR1-GFP* seedlings were treated for 6 hr with 0.5 mM SA followed by

*Figure 6 continued on next page*

*Figure 6 continued*

addition of 100 μM MG132 for a further 18 hr. Polyubiquitinated NPR1-GFP protein was then purified with GFP-Trap agarose and incubated for the indicated times with recombinant UBP6 in presence or absence of 26S proteasomes. Remaining polyubiquitinated NPR1-GFP and released ubiquitin species were detected by immunoblotting using an antibody against ubiquitin (P4D1), while unmodified NPR1-GFP was detected with an anti-GFP antibody. (D) *35S::NPR1-GFP* seedlings were treated for 2 hr with 0.5 mM SA followed by addition of 50 μM WP1130 or DMSO vehicle for a further 4 hr. Ubiquitinated proteins were pulled down using GST-TUBEs. Input and ubiquitinated NPR1-GFP (NPR1-Ub$_n$) were detected by immunoblotting with a GFP antibody. (E) Seedlings were treated with SA for 24 hr to induce NPR1 before addition of 100 μM CHX. Endogenous NPR1 protein levels were monitored by immunoblotting and GAPDH levels confirmed equal loading. (F) *35S::NPR1-GFP* seedlings were treated for 2 hr with 0.5 mM SA followed by addition of 50 μM WP1130 or DMSO vehicle for a further 4 hr. NPR1-GFP binding to the *as-1* motif of the *PR1* promoter element was quantified by ChIP with *npr1* seedlings serving as a negative control. Data points represent mean ± SD while letters denote statistically significant differences between samples (Tukey Kramer ANOVA; α = 0.05, n = 3). (G) Plants of the stated genotypes were treated with 0.5 mM SA for 24 hr before the expression of NPR1 target genes was analysed by qPCR. Data points represent mean ± SD while letters denote statistically significant differences between samples (Tukey Kramer ANOVA; α = 0.05, n = 3). (H) Plants were treated with or without 0.5 mM SA 24 hr prior to inoculation with $5 \times 10^6$ colony forming units (cfu)/ml *Psm* ES4326. Leaf discs were analysed for bacterial growth at three dpi. Error bars represent 95% confidence limits, while letters denote statistically significant differences between samples (Tukey Kramer ANOVA; α = 0.05, n = 8).

DOI: https://doi.org/10.7554/eLife.47005.015

The following figure supplement is available for figure 6:

**Figure supplement 1.** Deubiquitinating activity of UBP6.

DOI: https://doi.org/10.7554/eLife.47005.016

UBP6 has proteasome-activated DUB activity. Moreover, addition of WP1130 inhibitor completely blocked HA-UbVS labelling (*Figure 6B*), illustrating the effectiveness of this inhibitor on *Arabidopsis* UBP6.

To examine if UBP6 can cleave ubiquitin chains we incubated recombinant UBP6 with free ubiquitin chains or with di-ubiquitin of different linkage types and compared it to activity of recombinant human USP14. Similar to human USP14, *Arabidopsis* UBP6 displayed very little deubiquitination activity on free ubiquitin chains or di-ubiquitin of K48 and K63 linkage types (*Figure 6—figure supplement 1A–1C*). Only wild-type UBP6 but not UBP6(C113S) in which the catalytic cysteine residue was mutated, was weakly capable of trimming K63-linked chains in presence of 26S proteasomes, although this activity required very long incubation times (*Figure 6—figure supplement 1B*). These findings mirror the poor in vitro activity of human USP14 on free ubiquitin chains (*Lee et al., 2016*). Instead, human USP14 deubiquitinates anchored ubiquitin chains of various linkage types, including K48 linkages that target proteins for proteasome-mediated degradation (*Lee et al., 2016*). Therefore we proceeded to investigate if UBP6 activity cleaves ubiquitin chains anchored to NPR1. Indeed, incubation of purified polyubiquitinated NPR1-GFP with recombinant UBP6 and 26S proteasomes led to the release of ubiquitin conjugates of approximately hexa-ubiquitin chain length (*Figure 6C*). This release of ubiquitin conjugates was dependent on SA treatment, suggesting that UBP6 counteracts SA-induced polyubiquitination of NPR1 (*Figure 6—figure supplement 1D*). Together, these results demonstrate that UBP6 is an active DUB capable of removing ubiquitin chains en bloc from NPR1.

## Deubiquitination by UBP6 and UBP7 regulates NPR1 stability and transcriptional activity

So what is the effect of UBP6- and UBP7-mediated deubiquitination on NPR1 function? We found that treatment of *NPR1-GFP* (in *npr1*) seedlings with WP1130 inhibitor increased the levels of SA-induced polyubiquitinated NPR1-GFP while reducing the unmodified amount of this protein (*Figure 6D*). This suggests that UBP6 and UBP7 activities are required for deubiquitination of SA-induced NPR1-GFP, thereby rescuing it from degradation. To further examine this possibility, we analysed the stability of endogenous NPR1 protein in SA-treated *ubp6 ubp7* double mutants. CHX chase experiments revealed that compared to WT plants, NPR1 was destabilised in *ubp6 ubp7* mutants (*Figure 6E*). Accordingly, expression of FLAG-tagged UBP6 in the *ubp6/7* mutant background restored NPR1 stability (*Figure 6—figure supplement 1E*). These results demonstrate that UBP6 and UBP7 serve to stabilise NPR1 by removing ubiquitin chains that signal for its proteasome-mediated degradation.

Given the importance of progressive ubiquitination for the transcriptional activity of NPR1, we explored how UBP6- and UBP7-mediated deubiquitination might affect NPR1 coactivator function. Because UBP6 and UBP7 were required for NPR1-dependent *PR1* gene expression (*Figure 5D*), we questioned if NPR1 was still associated with the *PR1* promoter in absence of UBP6 and UBP7 activities. Surprisingly, ChIP experiments showed that SA-induced association of NPR1-GFP with the *PR1* promoter was strongly enhanced in presence of WP1130 inhibitor (*Figure 6F*). This suggests that UBP6 and UBP7 prevent the build-up of long polyubiquitin chains that block the transcriptional activity of NPR1. It also implies that similar to their yeast homologue, UBP6 and UBP7 exhibit proteasome inhibitory activity (*Hanna et al., 2006*). This activity is thought to delay degradation of proteasome substrates, thereby creating a window of opportunity for DUBs to deubiquitinate substrates and pardon them from proteolysis. Importantly, proteasome inhibitory activity does not require the catalytic active site (*Hanna et al., 2006*). Thus, to investigate how deubiquitination and proteasome inhibitory activities of UBP6 contribute to the regulation of NPR1 coactivator activity, we expressed FLAG-tagged wild-type UBP6 (FLAG-UBP6) and catalytically inactive UBP6(C113S) (FLAG-UBP6m) in *ubp6 ubp7* double mutants. While *ubp6 ubp7* mutants were compromised in SA-induced activation of all NPR1 target genes tested, expression of FLAG-UBP6 fully restored SA-responsiveness (*Figure 6G*). By contrast, FLAG-UBP6m restored SA-induced expression of only a subset, but not all NPR1 target genes. A distinction was observed between *WRKY* and *PR* genes, with the former requiring catalytic DUB activity of UBP6 while the latter did not (*Figure 6G*). These data indicate that catalytic and non-catalytic activities of UBP6 regulate distinct NPR1-dependent gene sets.

Finally we examined what the relevance is of UBP6- and UBP7-regulated transcriptional activity of NPR1 in context of plant immunity. We first treated plants with or without SA before challenge inoculation with virulent *Psm* ES4326. SA treatment induced resistance in WT plants but did not block bacterial propagation in *ubp6/7* plants (*Figure 6H*). Collectively, these data clearly demonstrate that UBP6 and UBP7 are required for NPR1 coactivator activity and associated development of SA-dependent immunity.

## Discussion

The ubiquitin-mediated proteasome system plays vital roles in the regulation of eukaryotic gene expression, in large part by controlling the abundance of transcriptional regulators. Paradoxically, proteasome-dependent instability of selected potent eukaryotic transcriptional activators is necessary for the expression of their target genes. It is thought that their transcription-coupled degradation ensures the target promoter is continuously supplied with fresh activators that reinitiate transcription, thereby maximising gene expression (*Geng et al., 2012*; *Kodadek et al., 2006*). However, this sacrificial process is energy-expensive (*Collins and Goldberg, 2017*; *Peth et al., 2013*), raising a dilemma of why such mechanisms evolved to regulate transcriptional activators. Our study on the immune coactivator NPR1, however, indicates that ubiquitin chain extension and trimming activities can fine-tune transcriptional outputs of unstable eukaryotic activators without strict requirement for sacrificial turnover.

We discovered that ubiquitination of NPR1 is a stepwise event, requiring the actions of CRL3 and the E4 ligase UBE4. In resting cells, CRL3-mediated turnover of NPR1 is important for preventing autoimmunity in absence of pathogen threat (*Spoel et al., 2009*). The NPR1-dependent autoimmune phenotype of *ube4* mutants is reminiscent of that observed in *cul3a cul3b* mutants (*Figure 1*) (*Spoel et al., 2009*), suggesting that in addition to CRL3 ligase, UBE4 is required to clear NPR1 from the nucleus and prevent untimely activation of immunity. In presence of SA, however, CRL3-mediated ubiquitination induced NPR1 coactivator activity, whereas formation of polyubiquitin chains by UBE4 blocked its activity and ultimately led to proteasome-mediated turnover (*Figures 2* and *3*). Rather than initiating substrate ubiquitination, E4 ligases are thought to extend existing ubiquitin chains (*Crosas et al., 2006*; *Koegl et al., 1999*), thereby determining substrate commitment to proteasome-mediated degradation and contributing to proteasome processivity (*Aviram and Kornitzer, 2010*; *Koegl et al., 1999*). Functionally these enzymes are emerging as important players in limiting the activity of NLR immune receptors as well as potent eukaryotic transcriptional regulators. *Arabidopsis* UBE4/MUSE3 works in concert with a CRL1/SCF^CPR1 ligase to regulate stability of the intercellular immune receptors SNC1 and RPS2 that recognise pathogen

invasion (*Cheng et al., 2011*; *Gou et al., 2012*; *Huang et al., 2014*). Taken together with our finding that UBE4 acts in concert with CRL3 (*Figure 3* and *Figure 3—figure supplement 1*), this suggests a single E4 enzyme may assist in diverse ubiquitin-mediated pathways controlled by different E3 ligases.

The role of *Arabidopsis* UBE4 in ubiquitination and degradation of NPR1 are reminiscent of stepwise ubiquitination of the mammalian tumour suppressor p53, a potent transcriptional activator of genes involved in apoptosis, cell cycle arrest and cellular senescence. The stability of p53 is regulated by amongst others the E3 ligase MDM2 (or HDM2 in humans) (*Pant and Lozano, 2014*). Although MDM2 limits p53 activity by promoting its turnover, MDM2 only catalyses multi-monoubiquitination of p53, which is insufficient for recognition by the proteasome (*Lai et al., 2001*). Progression to the polyubiquitinated form of p53 in the nucleus is carried out by the U-box E4 ligase UBE4B that interacts with both MDM2 and p53 (*Li et al., 2003*; *Wu and Leng, 2011*; *Wu et al., 2011*). Although this is similar to the proposed roles of CRL3 and UBE4 in controlling NPR1 stability, initial ubiquitination has different effects on p53 and NPR1. While MDM2-mediated monoubiquitination controls nucleocytoplasmic trafficking of p53 (*Li et al., 2003*), it probably does not have a direct effect on intrinsic p53 activator activity. Instead, initial ubiquitination boosts NPR1 transcriptional coactivator activity, at least in part by enhancing target promoter occupancy in the short term and potentially also by promoting genomic mobility of NPR1 in the longer term (*Figure 3*). Although it remains unclear if CRL3 adds only monoubiquitin or generates short chains shy of tetraubiquitin, the minimal signal required for proteasome recognition (*Thrower et al., 2000*), progression to polyubiquitin chain formation by UBE4 results in transcriptional shut down as polyubiquitinated NPR1 still occupied target promoters but lacked transcriptional potency (*Figure 6D and F*). This type of stepwise ubiquitination may be a general mechanism to control unstable transcriptional (co)activators in eukaryotes. For example, multi-monoubiquitination of the oncogenic growth coactivator SRC-3 results in its transcriptional activation, while subsequent chain extension targets it for degradation, but E4 ligases have not yet been implicated. We propose here that stepwise ubiquitination established by the sequential actions of E3 and E4 ligases may generate a transcriptional timer that controls the activity and lifetime of unstable (co)activators (*Figure 7*).

The complexity of the ubiquitin-dependent transcriptional timer was further revealed by the identification of UBP6 and UBP7 that deubiquitinated NPR1, thereby regulating its transcriptional activity and lifetime (*Figures 5* and *6*). Several unstable mammalian transcription activators, including p53 and the immune activator NF-κB, are also regulated by diverse DUBs (*Colleran et al., 2013*; *Pant and Lozano, 2014*; *Schweitzer and Naumann, 2015*). In these cases DUBs promote transcription by stabilising p53 and NF-κB at their target promoters. For example, loss of USP7-mediated deubiquitination of NF-κB resulted in increased turnover and decreased promoter occupancy of NF-κB (*Colleran et al., 2013*). Similarly, we found that knockout of UBP6 and UBP7 resulted in enhanced turnover and decreased transcriptional output of NPR1 (*Figure 6*). However, inhibition of UBP6/7 deubiquitination activities with WP1130 resulted in enhanced occupancy of transcriptionally inactive NPR1 at the *PR1* target promoter (*Figure 6F*). These data suggest that (*i*) like their yeast and mammalian counterparts (*Hanna et al., 2006*; *Lee et al., 2016*), UBP6 and UBP7 exhibit proteasome inhibitory activities that at least temporarily prolong promoter occupancy by NPR1, and (*ii*) UBP6 and UBP7 prevent inactivation of NPR1 by opposing the formation of long ubiquitin chains.

UBP6 showed a similar DUB activity as its mammalian homologue USP14 (*Lee et al., 2016*), in that it appeared to deubiquitinate NPR1 by removing ubiquitin chains en bloc (*Figure 6C and D*). Such activity places this DUB in direct opposition to UBE4-mediated chain extension activity. In yeast Ubp6 was previously reported to oppose ubiquitin chain extension activity of the E4 ligase Hul5, thereby regulating substrate recruitment to the proteasome (*Crosas et al., 2006*). Similarly, Arabidopsis UBP6 and UBP7 opposed ubiquitin ligase activities to extend the lifetime of transcriptionally active NPR1. Although we cannot completely rule out that these DUBs function in opposition to CRL3, their en bloc ubiquitin removal activity suggests they more likely remove longer ubiquitin chains generated by UBE4 (*Figure 7*).

Other DUBs may also play a role in regulating SA-responsive gene expression. This is illustrated by our findings that all tested DUB inhibitors blocked NPR1 target gene expression, while mutation of exclusively UBP6 and UBP7 had a similar effect. This discrepancy is expected because the specificity of DUB inhibitors has well-recognised limitations in that they often target multiple related DUBs (*Figure 5—figure supplement 1A*). This is likely especially true for *Arabidopsis thaliana* which

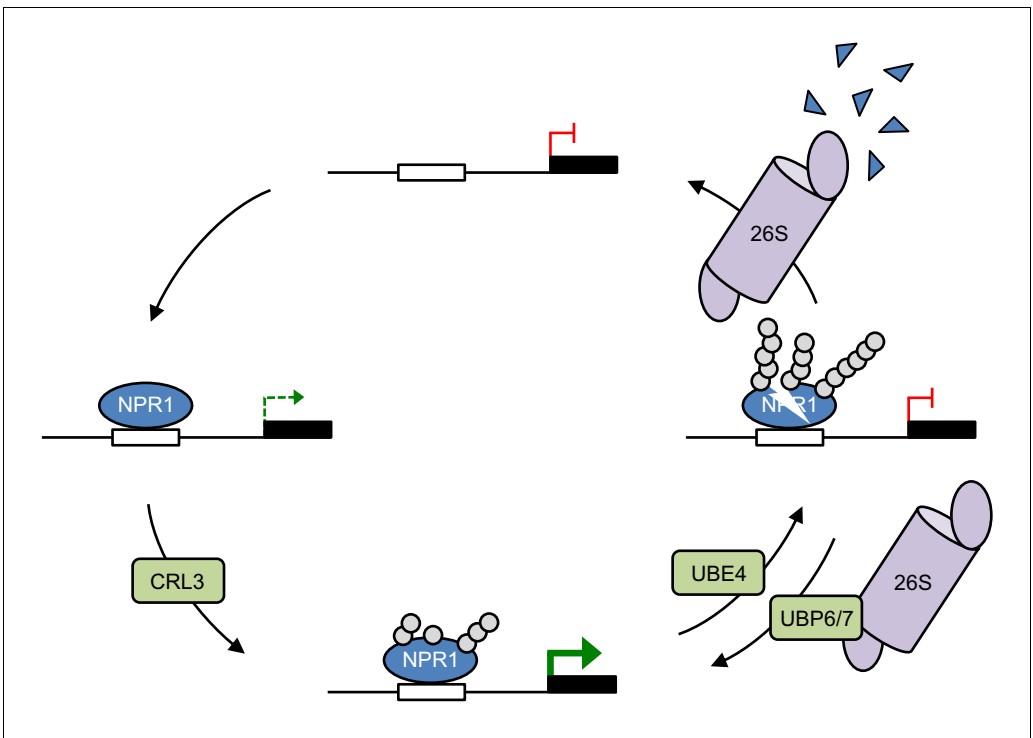

**Figure 7.** Working model for how dynamic ubiquitination regulates transcriptional outputs of NPR1. NPR1 occupancy at target gene promoters initiates low-level transcription (dashed green arrow). Initial ubiquitin (grey circles) modifications mediated by CRL3 ligase enhances target gene expression to maximum levels (solid green arrow), while progression to long-chain polyubiquitination mediated by UBE4 promotes the proteasome-mediated degradation of NPR1 and inactivates target gene expression. UBP6/7 activity at the proteasome serves to limit the degradation of NPR1, thereby promoting its active state.
DOI: https://doi.org/10.7554/eLife.47005.017

encodes for >60 DUB enzymes, many of which cluster into related sub-groups (*Liu et al., 2008*; *March and Farrona, 2017*). Besides UBP6/7 other DUBs may therefore also play a role in regulating SA-responsive gene expression but their precise functions remain unknown.

In summary, we report that disparate ubiquitin modifying enzymes play important roles in establishment of plant immune responses. We demonstrate that the opposing actions of an E3 and E4 ligase pair and two DUBs can fine-tune transcriptional outputs of the unstable immune coactivator NPR1 without strict requirement for its sacrificial turnover. Dynamicity in conjugated ubiquitin chain length may be a powerful mechanism for controlling the activity of unstable eukaryotic (co)activators in general.

# Materials and methods

## Key resources table

| Reagent type (species) or resource | Designation | Source or reference | Identifiers | Additional information |
|---|---|---|---|---|
| Genetic reagent (*Arabidopsis thaliana*) | *cul3a cul3b* | (*Spoel et al., 2009*) | SALK_046638 SALK_098014 | |
| Genetic reagent (*Arabidopsis thaliana*) | *ics1/sid2-2* | (*Wildermuth et al., 2001*) | N/A | |
| Genetic reagent (*Arabidopsis thaliana*) | *ube4-2* | (*Sessions et al., 2002*) | SAIL_713_A12 | |

*Continued on next page*

*Continued*

| Reagent type (species) or resource | Designation | Source or reference | Identifiers | Additional information |
|---|---|---|---|---|
| Genetic reagent (*Arabidopsis thaliana*) | *npr1-1* | (*Cao et al., 1994*) | N/A | |
| Genetic reagent (*Arabidopsis thaliana*) | *npr1-0* | This paper, (*Alonso et al., 2003*) | SALK_204100 | |
| Genetic reagent (*Arabidopsis thaliana*) | *35S::NPR1-GFP npr1-1* | (*Kinkema et al., 2000*) | N/A | |
| Genetic reagent (*Arabidopsis thaliana*) | *ubp12-2w* | (*Cui et al., 2013*) | GABI_742C10 | |
| Genetic reagent (*Arabidopsis thaliana*) | *uch3-1* | This paper, (*Alonso et al., 2003*) | SALK_140823 | |
| Genetic reagent (*Arabidopsis thaliana*) | *ubp6-1* | This paper, (*Alonso et al., 2003*) | SALK_108832 | |
| Genetic reagent (*Arabidopsis thaliana*) | *ubp7-1* | This paper, (*Alonso et al., 2003*) | SALK_014223 | |
| Genetic reagent (*Arabidopsis thaliana*) | *35S::FLAG-UBP6 ubp6/7* | This paper | SALK_108832 SALK_014223 | |
| Antibody | Mouse monoclonal anti-GFP | Roche | Cat# 11814460001 | (1:1000 – 1:2000) |
| Antibody | Rabbit polyclonal anti-S5a | Abcam | Cat# ab60101 | (1:10000) |
| Antibody | Rabbit polyclonal anti-NPR1 | This paper | N/A | (1:1000) |
| Antibody | Rabbit polyclonal anti-GAPDH | Sigma-Aldrich | Cat# G9545 | (1:5000) |
| Antibody | Rabbit polyclonal anti-pS11/15 NPR1 | (*Spoel et al., 2009*) | N/A | (1:1000) |
| Antibody | Rabbit polyclonal anti-GFP (ChIP grade) | Abcam | Cat# ab290 | (1:500 for ChIP) |
| Antibody | Mouse monoclonal anti-Ubiquitin (FK2) | Millipore | Cat# 04–263 | (1:2000) |
| Antibody | Mouse monoclonal anti-FLAG M2 affinity gel | Sigma-Aldrich | Cat# A2220 | N/A |
| Antibody | Rabbit monoclonal anti-FLAG | Sigma-Aldrich | Cat# F7425 | (1:2000) |
| Antibody | Rabbit polyclonal anti-RPN6 | Upstate | Cat# 11814460001 | (1:2000) |
| Antibody | Mouse monoclonal anti-Ubiquitin (P4D1) | Santa Cruz Biotechnology | Cat# sc-8017 | (1:2000) |
| Antibody | Mouse monoclonal anti-HA | ThermoFisher | Cat# 26183 | (1:5000) |
| Antibody | Mouse monoclonal anti-T7 | Millipore | Cat# 69522 | (1:5000) |
| Recombinant DNA reagent | pENTR-D-TOPO | Invitrogen | Cat# K240020 | |
| Recombinant DNA reagent | pEarleyGate 202 | ABRC (*Earley et al., 2006*) | Cat# CD3-688 | |
| Recombinant DNA reagent | pGEX-6P-1 | GE Healthcare | Cat# 28-9546-48 | |
| Recombinant DNA reagent | pET28a | Novagen | Cat# 69865 | |
| Peptide, recombinant protein | USP2 Catalytic Domain | Boston Biochem | Cat# E-504 | |

*Continued on next page*

*Continued*

| Reagent type (species) or resource | Designation | Source or reference | Identifiers | Additional information |
|---|---|---|---|---|
| Peptide, recombinant protein | 26S Proteasome (Ub-VS treated) | Ubiquigent | Cat# 65-1020-010 | |
| Peptide, recombinant protein | Poly-ubiquitin (Ub3-7) K48-linked | Boston Biochem | Cat# UC-220 | |
| Peptide, recombinant protein | Poly-ubiquitin (Ub3-7) K63-linked | Boston Biochem | Cat# UC-320 | |
| Peptide, recombinant protein | Di-ubiquitin K48-linked | Boston Biochem | Cat# UC-200B | |
| Peptide, recombinant protein | Di-ubiquitin K63-linked | Boston Biochem | Cat# UC-300B | |
| Peptide, recombinant protein | HA-Ubiquitin -Vinyl sulfone | Boston Biochem | Cat# U-212 | |
| Commercial assay or kit | SuperScript II | Invitrogen | Cat# 18064014 | |
| Commercial assay or kit | QuikChange Site-Directed Mutagenesis Kit | Agilent | Cat# 200519 | |
| Commercial assay or kit | GFP-Trap A | Chromotek | Cat# gta-20 | |
| Chemical compound, drug | PR-619 | Abcam | Cat# ab144641 | |
| Chemical compound, drug | NSC632839 | Abcam | Cat# ab144599 | |
| Chemical compound, drug | WP1130 | Cayman Chemical | Cat# 15227 | |
| Chemical compound, drug | P2207 | LifeSensors | Cat# SI9699 | |
| Chemical compound, drug | TCID | LifeSensors | Cat# SI9679 | |
| Chemical compound, drug | MG132 | Cayman Chemical | Cat# 10012628 | |
| Software, algorithm | Strand NGS | Avadis | N/A | |

## Plant maintenance, transformation, chemical treatments and pathogen infection

All *Arabidopsis* plants used in this study were in the Columbia genetic background, with WT referring to wild-type Col-0 throughout. Plants were grown under long day conditions (16 hr photoperiod) on soil in controlled-environment growth chambers at 65% humidity and 22°C unless otherwise stated. Seeds were stratified at 4–8°C in darkness for 2 days before moving to growth chambers. Plants were grown in a soil mix composed of peat moss, vermiculite and sand at a ratio of 4:1:1 respectively, and illumination was provided by fluorescent tube lighting at an intensity of 70–100 µmol m$^{-2}$sec$^{-1}$. For experiments on seedlings, seeds were sterilised by washing in 100% ethanol for 2 mins before incubating in 50% household bleach for 20 mins. After removal of bleach, seeds were washed at least 3 times with sterile H$_2$O before use. Sterilised seeds were spotted on Murashige and Skoog agar media and stratified before placing under lighting conditions as above. All T-DNA insertion mutants used were genotyped by PCR using standard conditions with gene specific primers in combination with left-border primers specific to each mutant collection (*Supplementary file 1*).

The coding sequences of the *UBE4* (At5g15400) and *UBP6* (At1g51710) genes were amplified using Phusion polymerase (NEB) from WT *Arabidopsis* cDNA with the addition of CACC at the 5' end required for TOPO cloning. The PCR products were gel-purified and cloned in to the pENTR/D-TOPO vector (Invitrogen) according to manufacturers' instructions. The active site residue of UBP6 was then mutagenised to serine (C113S) using QuikChange Site-Directed Mutagenesis Kit according

to manufacturers' instructions. Genes were then recombined into pEarleyGate 104 and 202 plasmids by LR reaction (Invitrogen) as described previously (*Earley et al., 2006*) to generate *35S::YFP-UBE4*, *35S::FLAG-UBP6* and *35S::FLAG-UBP6(C113S)* constructs. These plasmids were used to transform protoplasts or to transform *Agrobacterium tumifaciens* strain GV3101 (pMP90) as described previously (*Kneeshaw et al., 2014*). After selection of positive Agrobacterium clones carrying the transgenes, approximately 6 week old flowering *ubp6/7* plants were transformed as previously described (*Clough and Bent, 1998*). Selection of transformants was performed by spraying 10 day old seedlings with 120 µg/l BASTA at least three times. Further confirmation of transformation was performed by immunoblotting. Segregation of BASTA resistance was analysed in the $T_2$ generation to confirm plants had single transgene insertions.

For SA treatments, adult plants were sprayed with, while seedlings were immersed in 0.5 mM SA or $H_2O$. CHX, MG132 and DUB inhibitors were all used to treat seedlings by immersion at the concentrations stated in respective figure legends. Vehicle controls consisted of DMSO at the appropriate concentration for each chemical used.

*Psm* ES4326 was grown in LB media supplemented with 10 mM $MgCl_2$ and 50 µg/ml streptomycin. Cultures were grown overnight then centrifuged at 4,000 rpm for 10 mins. Cells were resuspended in 10 mM $MgCl_2$ and absorbance was measured at 600 nm before necessary dilutions were made to adjust concentrations to those indicated in figure legends. Plants were infected by pressure infiltration with a syringe through the abaxial leaf surface. For measurement of bacterial growth, a single leaf disc per plant was cut from infected leaves at the stated dpi and ground in 10 mM $MgCl_2$. Serial dilutions were plated on LB supplemented with 10 mM $MgCl_2$ and 50 µg/ml streptomycin and colonies were counted after 2 days incubation at 30°C.

## RNA extraction, cDNA synthesis and qPCR

Leaf tissue or whole seedlings were frozen and ground to a fine powder in liquid nitrogen. Samples were homogenised in RNA extraction buffer (100 mM LiCl, 100 mM Tris pH 8, 10 mM EDTA, 1% SDS) before addition of an equal volume of phenol/chloroform/isoamylalcohol (25:24:1). The homogenate was vortexed and centrifuged at 13,000 rpm for 5 min. The aqueous phase was transferred to an equal volume of 24:1 chloroform/isoamylalcohol, vortexed and then centrifuged at 13,000 rpm for 5 min. This step was repeated once before the aqueous layer was added to a 1/3 vol of 8 M LiCl and incubated overnight at 4°C. The extract was then centrifuged at 13,000 rpm for 5 min at 4°C. The resulting pellet was washed with ice cold 70% ethanol then rehydrated and dissolved in 400 µl $H_2O$ for 30 min on ice. Finally, 40 µl of NaAc (pH 5.3) and 1 ml of ice cold 96% ethanol was added before incubating for 1 hr at −20°C. The precipitate was then centrifuged at 13,000 rpm for 5 min at 4°C, the pellet was washed with ice cold 70% ethanol and resuspended in 50 µl of $H_2O$. Before cDNA synthesis, RNA samples were quantified using a NanoDrop spectrophotometer (Thermo Scientific) and appropriate dilutions were made to ensure all samples contained equal amounts of RNA. Reverse transcription was then performed using SuperScript II reverse transcriptase (Invitrogen) according to the manufacturers' instructions. qPCR was carried out on 20-fold diluted cDNA using Power SYBR Green (Life Technologies) and gene-specific primers (*Supplementary file 1*) on a StepOne Plus Real Time PCR machine (Life Technologies).

## RNA-Seq

RNA was extracted from biological duplicate samples as described above and further purified using an RNeasy Mini Kit (Qiagen) according to the manufacturer's instructions. qPCR was carried out to confirm appropriate induction of SA-responsive marker genes. RNA was then quantified and submitted to GATC Biotech/Eurofins (Constance, Germany) for RNA sequencing. The RNA-Seq reads were aligned to the *Arabidopsis thaliana* TAIR10 genome using Bowtie. TopHat identified potential exon-exon splice junctions of the initial alignment. Strand NGS software in RNA-Seq workflow was used to quantify transcripts. Raw counts were normalised using DESeq with baseline transformation to the median of all samples. Data were then expressed as normalised signal values (*i.e.* $\log_2$[RPKM] where RPKM is read count per kilobase of exon model per million reads) for all statistical tests and plotting. Genes were then filtered by expression (20%–100%) and differentially expressed genes determined by Benjamini Hochberg FDR with 2-way ANOVA (p=0.05). Additionally, we required SA-induced genes to meet a $\geq 2$ fold change cut-off, whereas NPR1-dependent genes required $\geq 1.5$ fold

change in Col-0 or *ube4* plants when compared to *npr1* mutants. RNA Seq data have been deposited in Array Express at EMBL-EBIunder accession code E-MTAB-7369.

## Chromatin immunoprecipitation

Chromatin immunoprecipitation was performed on leaf tissue of 4 week-old soil-grown adult plants essentially as described (*Yamaguchi et al., 2014*) but with minor modifications. 500 mg tissue was crosslinked with 1% formaldehyde by vacuum infiltration for 30 mins at room temperature. Glycine was added to a final concentration of 100 mM to quench crosslinking and vacuum infiltrated for a further 10 mins. Crosslinked tissue was washed twice with ice-cold PBS before all liquid was removed and tissue was frozen in liquid nitrogen. Nuclei were isolated and lysed as described (*Yamaguchi et al., 2014*) while sonication was performed using a BioRuptor Plus (Diagenode). Sonication consisted of 15 cycles of 30 s ON, 30 s OFF at high power. NPR1-GFP was immunoprecipitated using ChIP grade anti-GFP (Abcam) before capture of immune complexes with Protein A agarose (Millipore). Crosslink reversal and protein removal was performed as described previously (*Nelson et al., 2006*), by boiling in the presence of Chelex 100 resin (BioRad) before incubation at 55°C with Proteinase K. Finally, DNA was cleaned up using PCR purification columns (Qiagen) and analysed by qPCR using primers listed in *Supplementary file 1*.

## Protein analysis

For protein degradation assays and analysis of NPR1 levels, seedlings were frozen and ground to a fine powder in liquid nitrogen before homogenising in protein extraction buffer (PEB) (50 mM Tris-HCl (pH 7.5), 150 mM NaCl, 5 mM EDTA, 0.1% Triton X-100, 0.2% Nonidet P-40, and inhibitors: 50 μg/ml TPCK, 50 μg/ml TLCK, 0.6 mM PMSF) (*Spoel et al., 2009*). For analyses of NPR1 phosphorylation PEB buffer was supplemented with 1X phosphatase inhibitor cocktail 3 (Sigma). Samples were centrifuged at 13,000 rpm for 15 min at 4°C to clarify extracts, and the resulting supernatant was used for SDS-PAGE and immunoblot analysis. All antibodies used are listed in the Key Resources Table.

For analysis of polyubiquitination with TUBEs, seedlings were ground to a fine powder in liquid nitrogen and homogenised in 1x PBS supplemented with 1% Triton X-100, 10 mM NEM, 40 μM MG132, 50 μg/ml TPCK, 50 μg/ml TLCK, 0.6 mM PMSF, and 0.2 mg/ml GST-TUBE (*Hjerpe et al., 2009*). Homogenates were centrifuged at 13,000 rpm at 4°C for 20 mins to remove cellular debris and filtered through 0.22 μm filters before overnight incubation with Protino Glutathione Agarose 4B (Machery Nagel), at 4°C with rotation. The agarose beads were washed 5 times with 1X PBS + 1% Triton X-100 before elution by boiling in 1X SDS-PAGE sample buffer including 50 mM DTT. NPR1-GFP was detected by immunoblotting with anti-GFP (Roche).

For analysis of long chain polyubiquitination, seedlings were ground to a fine powder in liquid nitrogen and homogenised in 1X PBS, supplemented with 1% Triton X-100, 10 mM NEM, 80 μM MG115, 50 μg/ml TPCK, 50 μg/ml TLCK, 0.6 mM PMSF, 1X phosphatase inhibitor cocktail 3 (Sigma). Homogenates were centrifuged at 13,000 rpm at 4°C for 20 mins to remove cellular debris and filtered through 0.22 μm filters before overnight incubation with 300 μg $His_6$-V5-S5aUIM protein immobilised on agarose. Agarose beads were washed 5 times with extraction buffer before elution at 80°C for 15 mins in 1X SDS-PAGE sample buffer including 50 mM DTT. NPR1-GFP was detected by immunoblotting with anti-GFP (Roche).

For proteasome co-immunoprecipitation with FLAG-UBP6, seedlings were frozen and ground to a fine powder in liquid nitrogen before homogenising in proteasome extraction buffer (50 mM Tris-HCl (pH 7.4), 25 mM NaCl, 2 mM MgCl2, 1 mM EDTA, 10 mM ATP, 5% glycerol, and inhibitors: 50 μg/ml TPCK, 50 μg/ml TLCK, 0.6 mM PMSF). Extracts were centrifuged at 13,000 rpm at 4°C for 20 mins to remove cellular debris and filtered through 0.22 μm filters. Anti-FLAG M2 affinity gel was washed with the above buffer before incubating with samples overnight with rotation at 4°C. The resin was washed 3 times with the same buffer before immunoprecipitated proteins were eluted by boiling in 1X SDS-PAGE sample buffer including 50 mM DTT. FLAG-UBP6 was detected using rabbit anti-FLAG antibodies while co-immunoprecipitating proteins were detected with indicated antibodies.

## Recombinant protein and NPR1 antibody production

N-terminal GST-tagged TUBE was generated by cloning the coding sequence of hHR23A into pGEX-6P-1 using EcoRI and SalI restriction sites. Primers used are listed in *Supplementary file 1*. GST-TUBE expression was induced in BL21(DE3) *E. coli* cells with the addition of 1 mM IPTG and cultures were incubated for a further 4 hr at 28°C before collecting by centrifugation. Cells were then lysed in 1X PBS supplemented with 1 mg/ml lysozyme, 25 U/ml Benzonase nuclease, 0.1% Triton-X-100 and a protease inhibitor cocktail before GST-TUBE was purified using Protino Glutathione Agarose 4B according to the manufacturers' instructions. Purified GST-TUBE was dialysed against 1X PBS and stored with the addition of 10% glycerol at −80°C until use.

Recombinant S5aUIM protein was generated by synthesising residues 196–309 from human S5a with codon optimisation for *E. coli* into pET151/D-TOPO. The resulting $His_6$-V5-S5aUIM protein was expressed in BL21(DE3) *E. coli* cells by addition of 1 mM IPTG and incubation for 24 hr at 28°C before collecting by centrifugation. Cells were then lysed in lysis buffer (50 mM KHPO4 pH 8, 100 mM NaCl, 10 mM Imidazole, 1X BugBuster (Merck), 25 U/ml Benzonase nuclease, 50 µg/ml TPCK, 50 µg/ml TLCK and 0.5 mM PMSF). $His_6$-UBP6 was then purified using HisPur cobalt resin (Thermo Fisher) according to manufacturers' instructions. Purified $His_6$-V5-S5aUIM was dialysed against 1X PBS and covalently coupled to NHS-activated agarose to a final concentration of approximately 10 µg/µl following the manufacturer's instructions (Thermo Fisher).

N-terminal $His_6$-T7-tagged UBP6 was generated by cloning the coding sequence of *Arabidopsis UBP6* in to the expression vector pET28a using EcoRI and SalI restriction sites. Primers used are listed in *Supplementary file 1*. Expression was induced in BL21(DE3) *E. coli* cells with the addition of 1 mM IPTG and cultures were incubated for a further 3 hr at 28°C before collecting by centrifugation. Cells were then lysed in lysis buffer (50 mM KHPO4 pH 8, 300 mM NaCl, 10 mM Imidazole, 1 mg/ml lysozyme, 25 U/ml Benzonase nuclease, 0.1% Triton-X- 100, 10 mM β-mercaptoethanol, 50 µg/ml TPCK, 50 µg/ml TLCK and 0.5 mM PMSF). $His_6$-UBP6 was then purified using HisPur cobalt resin (Thermo Fisher) according to manufacturers' instructions. Purified $His_6$-UBP6 was dialysed against 50 mM Tris-HCl pH 7.4, 5M NaCl and stored with the addition of 10% glycerol at −80°C until use.

Recombinant FLAG-UBE4 was produced using cell-free synthesis via two-step PCR (*Nomoto and Tada, 2018*) using the primers listed in *Supplementary file 1*.

The anti-NPR1 polyclonal antibody was generated by immunising rabbits with a synthetic peptide based on a region of the NPR1 protein with the sequence N'-SALAAAKKEKDSNNTAAVKL-Cys. Rabbits were subsequently bled and antibodies were enriched by affinity purification (Proteintech, USA).

## HA-UbVS labelling and in vitro deubiquitination assays

For HA-UbVS labelling, 10 µl reactions were prepared in 50 mM Tris-HCl pH 7.4, 5 mM $MgCl_2$, 1 mM DTT and 1 mM ATP. Before labelling, 350 nM $His_6$-T7-UBP6 was pre-incubated with 50 µM WP1130 or DMSO control for 10 mins before addition of 10 nM Ub-VS treated 26S proteasomes (Ubiquigent). Reactions were incubated for a further 20 mins before addition of 700 nM HA-UbVS and further incubation for 30 mins. All steps were carried out at room temperature. Labelling was terminated with the addition of SDS-PAGE sample buffer including 50 mM DTT. Samples were heated at 70°C for 10 mins before SDS-PAGE and immunoblot analyses.

All in vitro deubiquitination assays were performed in DUB buffer (50 mM Tris-HCl pH 7.4, 5 mM $MgCl_2$, 1 mM DTT, 5 mM ATP). Where indicated, 1.25 nM Ub-VS treated 26S proteasomes and 20 nM UBP6 were added. Di-ubiquitin and polyubiquitin chain substrates were included at 400 nM. Reactions were incubated at 30°C for the times indicated in figure legends before terminating with addition of SDS-PAGE sample buffer including 50 mM DTT. Samples were heated at 70°C for 10 mins before SDS-PAGE and immunoblot analyses.

For in vitro deubiquitination of NPR1-GFP isolated from plants, seedlings were treated with SA and MG132 as described in figure legends. Seedlings were frozen and ground to a fine powder in liquid nitrogen before homogenising in protein extraction buffer (PEB) (50 mM Tris-HCl (pH 7.5), 150 mM NaCl, 5 mM EDTA, 0.1% Triton X-100, 0.2% Nonidet P-40, and inhibitors: 50 µg/ml TPCK, 50 µg/ml TLCK, 0.6 mM PMSF). Extracts were centrifuged at 13,000 rpm at 4°C for 20 mins to remove cellular debris and filtered through 0.22 µm filters. GFP-Trap A agarose (Chromotek) was incubated with extracts for 2 hr with rotation at 4°C before washing 10 times with PEB (without inhibitors) then twice with DUB buffer. Supernatant was completely removed before DUB reactions were set up as

described above but with NPR1-GFP immobilised on GFP-Trap A as the substrate. Proteins were eluted by boiling in 1X SDS-PAGE sample buffer including 50 mM DTT, before analysis by immunoblotting.

## Quantification and statistical analyses

For pathogen growth experiments confidence intervals that would allow acceptance or rejection of the null hypothesis were used to estimate sample size, while no statistical methods were used to pre-determine sample sizes elsewhere, nor were any methods of randomization. All experiments were repeated a minimum of two times with similar results. For quantitative immunoblotting, band intensities were acquired using Image Studio software and a LI-COR Odyssey FC imaging device (LI-COR Biosciences). NPR1 signal was normalised to GAPDH signal for three replicates, each of which consisted of an independent pool of at least 50 seedlings. For qPCR and ChIP-qPCR experiments with adult plants, samples consisted of a pool of at least six different plants while for seedlings, samples consisted of a pool of at least 50 seedlings. qPCR data is shown for three technical replicates of a representative repeated experiment. For pathogen growth experiments, replicates represent eight separate plants. In all figure legends, the statistical tests applied are stated while *n* refers to sample size.

## Acknowledgements

We thank Dr. Xin Li and Dr Xia Cui for sharing *muse3* and *ubp12-2w* seeds, respectively. This work was supported by a Royal Society University Research Fellowship (UF090321), a BBSRC grant (BB/L006219/1), and the European Research Council (ERC) under the European Union's Horizon 2020 research and innovation programme (grant agreement No 678511).

## Additional information

### Funding

| Funder | Grant reference number | Author |
| --- | --- | --- |
| Royal Society | UF090321 | Steven H Spoel |
| Biotechnology and Biological Sciences Research Council | BB/L006219/1 | Steven H Spoel |
| H2020 European Research Council | 678511 | Steven H Spoel |

The funders had no role in study design, data collection and interpretation, or the decision to submit the work for publication.

### Author contributions

Michael J Skelly, Conceptualization, Formal analysis, Supervision, Investigation, Visualization, Methodology, Writing—original draft, Writing—review and editing; James J Furniss, Heather Grey, Ka-Wing Wong, Investigation; Steven H Spoel, Conceptualization, Supervision, Funding acquisition, Investigation, Visualization, Methodology, Writing—original draft, Project administration, Writing—review and editing

### Author ORCIDs

Michael J Skelly (iD) http://orcid.org/0000-0002-9024-0037
Steven H Spoel (iD) https://orcid.org/0000-0003-4340-7591

### Decision letter and Author response

Decision letter https://doi.org/10.7554/eLife.47005.023
Author response https://doi.org/10.7554/eLife.47005.024

## Additional files

### Supplementary files
• Supplementary file 1. List of oligonucleotides used.
DOI: https://doi.org/10.7554/eLife.47005.018

• Transparent reporting form DOI: https://doi.org/10.7554/eLife.47005.019

### Data availability
RNA Seq data have been deposited in Array Express at EMBL-EBI under accession code E-MTAB-7369.

The following dataset was generated:

| Author(s) | Year | Dataset title | Dataset URL | Database and Identifier |
|---|---|---|---|---|
| Skelly MJ, Furniss JJ, Grey HL, Wong KW, Spoel SH | 2019 | Salicylic acid-induced gene expression in wild-type Col-0 and mutant ube4 Arabidopsis thaliana plants | https://www.ebi.ac.uk/arrayexpress/experiments/E-MTAB-7369/ | ArrayExpress, E-MTAB-7369 |

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
