## [Decision Letter]

Thank you for submitting your article "Dynamic ubiquitination determines transcriptional activity of the plant immune coactivator NPR1" for consideration by *eLife*. Your article has been reviewed by three peer reviewers, one of whom is a member of our Board of Reviewing Editors, and the evaluation has been overseen by Christian Hardtke as the Senior Editor. The reviewers have opted to remain anonymous.

The reviewers have discussed the reviews with one another and the Reviewing Editor has drafted this decision to help you prepare a revised submission. We would like to emphasize that we are overall very positive about the work and would like to see it published in *eLife* after appropriate revision.

Summary:

This work uncovers highly complex and dynamic ubiquitination of NPR1, the master regulator of the SA pathway. CUL3, UBE4, and UBP6/7 act together to regulate progressive ubiquitination of NPR1, thereby controlling NPR1 activity, in addition to previously known ubiquitin dependent degradation. The work is of great interest to the broad field of transcriptional regulation, and is overall well executed.

Essential revisions:

1) In most cases, the authors have used only a single allele for T-DNA knock-out mutants (most important for *ube4* and *ubp6 ubp7*). Unless functional complementation is provided, this is not robust enough to come up with a solid conclusion. Please provide additional genetic evidence supporting the role of UBE4 and UBP6/7 in NPR1 ubiquitination using other alleles.

2) The role of UBE4 in promoting NPR1 degradation are not entirely convincing. To show that NPR1 stability is affected by UBE4, they generated two lines, one expressing NPR1-GFP in Col-0 or, the other in the *ube4* background. Unexpectedly, NPR1 either accumulates to lower, or similar levels in the mutant background, compared to WT (see Figures 2A and 2C respectively). Have the authors analysed the transcript levels in these lines, and are they comparable?

3) Because Figure 2A is central to the manuscript, it would be good to have some quantification here. The loading control is not really equal, protein concentration seems to go down in the Col-0, while up in the *ube4* background. Normalising the signal to the corresponding S5a would help to make the point. Also related, is S5a not degraded via the proteasome? If authors have the coomassie, it would be good if they would include it.

4) It will be necessary to test association between UBE4 and NPR1 in vivo on SA-treated plants to support a direct action of UBE4 on NPR1. Finally, the ability of UBE4 to elongate ubiquitin chains is not consistent with Figure 2D or 6C where the ubiquitin chains seem to have the same length (reaching the same size). To convince the reader, the authors must provide more resolutive blots for high molecular weight. The elongation or trimming activities of UBE4 and UBP6/7, respectively, should result in marked migration differences on a western blot.

5) UBP6 is proposed to trim ubiquitin off NPR1 upon SA treatment, counteracting UBE4 (Figure 6C). This conclusion would be more convincing if the authors performed the exact same experiment without SA treatment. One should expect not to see the 50kDa 6x ubiquitin band showing up if UBP6 acts only on UBE4-extended chains.

6) Since UBE4 is also required for the degradation of multiple NLRs. The elevated defense gene expression and disease resistance in the *ube4* mutant observed may be a combination of both NPR1 function and NLR function. Although the authors show that the elevated defense gene expression and increased disease resistance in *ube4* mutant require ICS1 and NPR1, the same maybe true for NLR activated defenses. This will need to be thoroughly discussed.

---

## [Author Response]

Essential revisions:1) In most cases, the authors have used only a single allele for T-DNA knock-out mutants (most important for ube4 and ubp6 ubp7). Unless functional complementation is provided, this is not robust enough to come up with a solid conclusion. Please provide additional genetic evidence supporting the role of UBE4 and UBP6/7 in NPR1 ubiquitination using other alleles.

We have now provided additional genetic evidence for *ube4* and *ubp6 ubp7*, which further support our conclusions on their roles in controlling NPR1 ubiquitination:

a) In addition to the *ube4-2*, we used the previously described *muse3* mutant allele of *UBE4* (Huang et al., 2014). New Figure 2—figure supplement 1D and E show that both mutant alleles compromise degradation of the endogenous SA-induced NPR1 protein.

Moreover, in new Figure 3—figure supplement 1A we show that like in *ube4-2* plants, SA-induced *PR1* gene expression is enhanced in the *muse3* mutant.

b) In the original manuscript we already complemented *ubp6 ubp7* mutants with a *35S::FLAG-UBP6* construct. We now show in new Figure 6—figure supplement 1E that enhanced NPR1 turnover observed in *ubp6 ubp7* mutants is indeed alleviated by complementation with *35S::FLAG-UBP6*. Furthermore, in Figure 6G and 6F we already demonstrate that impaired expression of NPR1 target genes in the *ubp6 ubp7* mutant is rescued to near WT levels when FLAG-UBP6 is expressed.

2) The role of UBE4 in promoting NPR1 degradation are not entirely convincing. To show that NPR1 stability is affected by UBE4, they generated two lines, one expressing NPR1-GFP in Col-0 or, the other in the ube4 background. Unexpectedly, NPR1 either accumulates to lower, or similar levels in the mutant background, compared to WT (see Figures 2A and 2C respectively). Have the authors analysed the transcript levels in these lines, and are they comparable?

Thank you for this suggestion. To accurately compare stability of the NPR1-GFP protein in different genetic backgrounds, we ensured that *NPR1-GFP* transcript levels were comparable (new Figure 2—figure supplement 1B). Nonetheless, accumulation of SA-induced NPR1-GFP protein is somewhat variable between experiments (see replica samples in new Figure 2—figure supplement 1C), which was mirrored by the endogenous NPR1 protein (see replica samples in new Figure 2—figure supplement 1D and E). Therefore we strictly used qualitative and quantitative cycloheximide (CHX) chase assays rather than steady-state levels to draw conclusions on the stability of NPR1. This well established proteostasis approach clearly shows that NPR1-GFP and endogenous NPR1 are both stabilised in *ube4* mutants.

3) Because Figure 2A is central to the manuscript, it would be good to have some quantification here. The loading control is not really equal, protein concentration seems to go down in the Col-0, while up in the ube4 background. Normalising the signal to the corresponding S5a would help to make the point. Also related, is S5a not degraded via the proteasome? If authors have the coomassie, it would be good if they would include it.

Quantification of our data and improving loading controls are excellent suggestions. We have done the following:

a) New Figure 2—figure supplements 1C-E show the quantification (n=3) of NPR1-GFP and endogenous NPR1 protein levels in CHX chase assays. These data demonstrate that NPR1 protein is significantly more stable in the *ube4* mutant background.

b) Rather than using the S5a protein, we have now used the more stable GAPDH protein to normalise the data. Thus, we repeated the experiment shown in new Figure 2A to include GAPDH as a loading control and we also utilised GAPDH to normalise the quantitative data on NPR1 stability shown in new Figure 2—figure supplements 1C-E.

4) It will be necessary to test association between UBE4 and NPR1 in vivo on SA-treated plants to support a direct action of UBE4 on NPR1. Finally, the ability of UBE4 to elongate ubiquitin chains is not consistent with Figure 2D or 6C where the ubiquitin chains seem to have the same length (reaching the same size). To convince the reader, the authors must provide more resolutive blots for high molecular weight. The elongation or trimming activities of UBE4 and UBP6/7, respectively, should result in marked migration differences on a western blot.

As an E4 ligase the UBE4 protein is not expected to participate in direct protein-protein interaction with NPR1 itself. Rather UBE4 would be expected to transiently bind to the ubiquitin chain that decorates NPR1. In accordance, we have found it difficult to repeatedly co-immunoprecipitate UBE4 and NPR1 in vivo, which is not unusual for highly transient enzyme-substrate interactions. Nonetheless, using highly sensitive mass spectrometry approaches we observed that in some but not all experiments UBE4 co-immunoprecipitated with NPR1-GFP in vivo. Encouraged by this result we then immunoprecipitated ubiquitinated NPR1-GFP from plants that were treated with SA and proteasome inhibitor. Ubiquitinated NPR1-GFP was then incubated with in vitro synthesised FLAG-UBE4 and interaction assessed by co-immunoprecipitation. To our delight new Figure 2F demonstrates that FLAG-UBE4 indeed associated with polyubiquitinated NPR1-GFP.

Regarding ubiquitin chain length, it should be noted that protein ubiquitination results in a branched (rather than linear) polypeptide, the size of which by definition cannot be determined accurately by SDS-PAGE. Indeed, it is known that even unconjugated, non-linear polyubiquitin chains do not migrate at expected molecular size. Our immunoblots are already performed on low percentage acrylamide gels so further resolution cannot be obtained, nor would it be a reliable indicator of ubiquitin chain length. For this very reason we performed pull down experiments with recombinant S5a ubiquitin interacting motifs (S5aUIM) that preferentially bind chains of four or more ubiquitin molecules (Deveraux et al., 1994; Young et al., 1998). This experiment shows that UBE4 promotes formation of long ubiquitin chains on NPR1 (see Figure 2E).

Moreover, differences in ubiquitination patterns observed in Figures 2D and 6C are due to distinct detection methods. While Figure 2D was visualised with an α-GFP antibody to detect only polyubiquitinated NPR1-GFP protein, Figure 6C was visualised with an α-ubiquitin antibody and thus shows both conjugated and free ubiquitin chains.

5) UBP6 is proposed to trim ubiquitin off NPR1 upon SA treatment, counteracting UBE4 (Figure 6C). This conclusion would be more convincing if the authors performed the exact same experiment without SA treatment. One should expect not to see the 50kDa 6x ubiquitin band showing up if UBP6 acts only on UBE4-extended chains.

This is a great suggestion. Indeed, new Figure 6—figure supplement 1Dshows that only after SA treatment UBP6 released smaller ubiquitin species, suggesting it primarily acts only UBE4 extended ubiquitin chains.

6) Since UBE4 is also required for the degradation of multiple NLRs. The elevated defense gene expression and disease resistance in the ube4 mutant observed may be a combination of both NPR1 function and NLR function. Although the authors show that the elevated defense gene expression and increased disease resistance in ube4 mutant require ICS1 and NPR1, the same maybe true for NLR activated defenses. This will need to be thoroughly discussed.

Good point. We have adjusted our conclusions from these data in the Results section. We now described that:

“In *Arabidopsis* the E4 ligase UBE4/MUSE3 has been implicated in the degradation of NLR (nucleotide binding and leucine-rich repeat) immune receptors. […] Therefore we investigated if UBE4 is involved in downstream NPR1-dependent immune signalling by acquiring a loss-of-function T-DNA insertion mutant (Figure 1—figure supplement 1).”

Our concluding remarks for this section have been altered to:

“Collectively, these data suggest that in unchallenged plants UBE4 suppresses the expression of SA-mediated NPR1 target genes and prevents autoimmunity, conceivably by altering the stability of upstream NLR immune receptors as well as the downstream NPR1 coactivator.”